# PRISM: PRIVACY-PRESERVING IMPROVED STOCHASTIC MASKING FOR FEDERATED GENERATIVE MODELS

**Kyeongkook Seo[1], Dong-Jun Han[2] \*, Jaejun Yoo[1]\***
[1]Ulsan National Institute of Science and Technology (UNIST)
[2]Department of Computer Science and Engineering, Yonsei University
{kyeongkookseo, jaejun.yoo}@unist.ac.kr, djh@yonsei.ac.kr

## ABSTRACT

Despite recent advancements in federated learning (FL), the integration of generative models into FL has been limited due to challenges such as high communication costs and unstable training in heterogeneous data environments. To address these issues, we propose PRISM, a FL framework tailored for generative models that ensures (i) stable performance in heterogeneous data distributions and (ii) resource efficiency in terms of communication cost and final model size. The key of our method is to search for an optimal stochastic binary mask for a random network rather than updating the model weights, identifying a sparse subnetwork with high generative performance; *i.e.*, a "strong lottery ticket". By communicating binary masks in a stochastic manner, PRISM minimizes communication overhead. Combined with the utilization of maximum mean discrepancy (MMD) loss and a mask-aware dynamic moving average aggregation method (MADA) on the server side, PRISM facilitates stable and strong generative capabilities by mitigating local divergence in FL scenarios. Moreover, thanks to its sparsifying characteristic, PRISM yields an lightweight model without extra pruning or quantization, making it ideal for environments such as edge devices. Experiments on MNIST, FMNIST, CelebA, and CIFAR10 demonstrate that PRISM outperforms existing methods, while maintaining privacy with minimal communication costs. PRISM is the first to successfully generate images under challenging non-IID and privacy-preserving FL environments on complex datasets, where previous methods have struggled. Our code is available at PRISM.

## 1 INTRODUCTION

Recent generative models have demonstrated remarkable advancements in image quality and have been widely extended to various domains, including image-to-image translation (Choi et al., 2020; Saharia et al., 2022)], layout generation (Seol et al., 2024)], text-to-image generation (Rombach et al., 2022; Ramesh et al., 2022)], and video generation (Skorokhodov et al., 2022; Kim et al., 2024b)]. Achieving high-quality generation with current generative models demands increasingly large datasets, leading to concerns that publicly available data will soon exhausted (Villalobos et al., 2024)]. Leveraging the vast amount of data stored on edge devices becomes a potential solution, but this poses significant challenges: not only does the private nature of the data make centralized training impractical, but edge computing itself faces hurdles, including limited resources and prohibitive communication costs.

Federated learning (FL) (McMahan et al., 2017)] is a promising paradigm tailored to this setup, enabling clients to collaboratively train a global model without sharing their local datasets with a third party. However, high communication costs, performance degradation due to data heterogeneity, and the need to preserve privacy remain significant challenges in FL. These challenges are further intensified in the context of generative models. Unlike classification tasks, generative tasks lack a well-defined objective function and focus on learning the data sample distribution, making the inte-

---

*Co-corresponding author.

gration of FL and generative models even more difficult. A few recent works have made efforts to train generative models over distributed clients (Hardy et al., 2019; Rasouli et al., 2020; Li et al., 2022; Zhang et al., 2021; Amalan et al., 2022)]. These methods are generally built upon generative adversarial networks (GANs) (Goodfellow et al., 2020)], which have shown impressive results in the field of image generation. DP-FedAvgGAN (Augenstein et al., 2019)], GS-WGAN (Chen et al., 2020)], and Private-FLGAN (Xin et al., 2020)] apply differential privacy (DP) (Dwork et al., 2006; Mironov, 2017)] to mitigate the potential privacy risk in FL setups. However, existing works still face several challenges: 1) Previous approaches (Farnia & Ozdaglar, 2020a;b; Wang et al., 2022)] underperform, especially in non-IID (independent, identically distributed) data distribution scenarios with strong data heterogeneity across FL clients due to the notorious instability of GANs. 2) Performance evaluations are limited to relatively simple datasets such as MNIST, Fashion MNIST, and EMNIST. 3) They suffer from significant communication overhead during model exchanges between the server and clients.

To overcome these challenges, we propose PRivacy-preserving Improved Stochastic-Masking for generative models (PRISM), a new strategy for training generative models in FL settings with the following key features: **First**, at the heart of PRISM is the strong lottery ticket (SLT) hypothesis (Frankle & Carbin, 2018)], suggesting the existence of a highly effective subnetwork within a randomly initialized network. PRISM shifts the focus towards identifying an optimal global binary mask, rather than updating the weights directly. This approach enables each client to transfer the binary mask to the server instead of the full model, significantly reducing the overload in each communication round. Moreover, when training is finished, PRISM produces a lightweight final model, as each weight is already quantized, thanks to our initialization strategy. This feature provides significant advantages for resource-constrained edge devices. **Second**, PRISM incorporates the maximum mean discrepancy (MMD) loss (Gretton et al., 2006; 2012)] during client-side updates, ensuring stable training for generative models. **Third**, a mask-aware dynamic moving average aggregation (MADA) is introduced to alleviate local model divergence. This allows PRISM to maintain the previous global mask information and alleviate client drift under non-IID and DP-guaranteeing scenarios. By automatically adjusting the moving average parameter based on mask correlations, this approach requires neither a regularization term nor hyperparameter tuning. **Finally,** PRISM offers a hybrid strategy that can flexibly trade-off between image quality and communication cost. Taken together, these features enable PRISM to consistently deliver robust performance in challenging non-IID and DP-guaranteeing FL settings, while maintaining minimal communication overhead.

Our experimental results reveal that PRISM sets a new standard in generative model performance, significantly outperforming GAN-based methods in both IID and non-IID scenarios. It acheives state-of-the-art image generation on complex datasets such as CelebA and CIFAR10, whereas previous methods were limited to simpler datasets like MNIST and FMNIST. This highlights PRISM's potential for scalable and resource-efficient generative model learning in distributed environments. Overall, our main contributions can be summarized as follows:

- We propose PRISM, an effective FL framework that achieves state-of-the-art performance on various benchmark datasets. It is the first method to successfully generate images on complex datasets such as CelebA in FL scenarios that involves data heterogeneity and privacy preservation.

- PRISM offers an efficient solution for federated generative models with minimal communication overhead by incorporating SLT with a stochastic binary mask. Even more, in conjunction with the weight initialization strategy, the final model acquired from PRISM becomes significantly lightweight, reducing to less than half the size of the initial model.

- We further enhance the stability of federated learning for generative models by introducing MMD loss and a mask-aware dynamic moving average aggregation method (MADA).

To the best of our knowledge, this is the first work to address the challenges in communication efficiency, privacy, stability, and generation performance altogether for federated generative models. We provide new directions to this area based on several unique characteristics, including SLT with stochastic binary mask, MMD loss, mask-aware dynamic moving average aggregation strategy, and hybrid score/mask communications.

## 2 RELATED WORK

**Federated learning for classification models.** FL has achieved a significant success in training a global model in a distributed setup, eliminating the necessity of sharing individual client's local datasets with either the server or other clients. Research has been conducted on various aspects of FL, such as data heterogeneity (Zhao et al., 2018; Li et al., 2021b)], communication efficiency (Isik et al., 2022; Li et al., 2021a; Mitchell et al., 2022; Basat et al., 2022)], privacy (Wei et al., 2020)], with most focusing on image classification tasks. Related to our approach, FedPM (Isik et al., 2022)] and FedMask (Li et al., 2021a)] adopted binary mask communication to reduce the communication costs in FL in classification tasks. FedMask (Li et al., 2021a)] introduces binary mask communication, focusing on communication efficiency and personalization in decentralized environments, while FedPM (Isik et al., 2022)] utilizes stochastic masks to minimize uplink overhead and proposes a bayesian aggregation method to robustly manage scenarios with partial client participation. While the concept of incorporating SLT into FL paradigm has been studied in FedMask (Li et al., 2021a)] and FedPM (Isik et al., 2022)] for *classification tasks*, PRISM stands as an independent strategy tailored for the training of *generative models* across distributed clients: PRISM incorporates MMD loss for more robust performance compared to GAN-based approaches and introduces MADA to maintain a stable image generation performance in heterogeneous and DP-guaranteeing FL settings.

**Federated learning for generative models.** Generative models, like classification models, also offer potential benefits in various federated settings (See Appendix P). Several recent works have aimed to incorporate generative models into distributed settings (Hardy et al., 2019; Amalan et al., 2022; Li et al., 2022; Zhang et al., 2021; Rasouli et al., 2020; Augenstein et al., 2019; Chen et al., 2020; Xin et al., 2020)]. MD-GAN (Hardy et al., 2019)] was the first attempt to apply generative models in the FL framework using GANs (Goodfellow et al., 2020)], extensively studied in image generation tasks. In MD-GAN, each client holds a discriminator, and the server aggregates the discriminator feedback from each client to train the global generator. To prevent overfitting of local discriminators, clients exchange discriminators, incurring additional communication costs. Multi-FLGAN (Amalan et al., 2022)] proposed all vs. all game approach by employing multiple generators and multiple discriminators and then selecting the most powerful network to enhance model performance. IFL-GAN (Li et al., 2022)] improves both performance and stability by weighting each client's feedback based on the MMD between the images generated by the global model and the local generator. This approach maintains a balance between the generator and the discriminator, leading to Nash Equilibrium. Other works such as (Zhang et al., 2021; Rasouli et al., 2020)] have also explored the utilization of GANs in FL. However, these works do not consider the challenge of privacy preservation in the context of FL and also suffer from resource issues during training and inference.

**Federated learning for generative models with privacy consideration.** Only a few prior works have focused on the privacy preservation in federated generative models (Augenstein et al., 2019; Chen et al., 2020)]. DP-FedAvgGAN (Augenstein et al., 2019)] introduces to combine federated generative models and differential privacy (DP) (Dwork et al., 2006; Mironov, 2017)] to ensure privacy preservation. GS-WGAN (Chen et al., 2020)] adopts Wasserstein GAN (Gulrajani et al., 2017)] to bypass the cumbersome searching for an appropriate DP-value, leveraging the Lipschitz property. While these approaches have successfully integrated FL and generative models, they inherit drawbacks such as the notorious instability of GANs (Farnia & Ozdaglar, 2020a;b; Wang et al., 2022)] and significant performance drop under data heterogeneity. Moreover, all existing approaches suffer from huge communication and storage costs during and after training, respectively.

## 3 BACKGROUND

### 3.1 STRONG LOTTERY TICKETS

Strong Lottery Ticket (SLT) hypothesis (Frankle & Carbin, 2018; Malach et al., 2020; Orseau et al., 2020)] suggest the existence of a sparse subnetwork within an initially random network that achieves a superior performance. Edge-Popup (EP) algorithm (Ramanujan et al., 2020)] is one of the most popular methods to discover SLT within the dense network, which introduces a scoring mechanism to select potentially important weights among the widespread initialized weight values. More specifically, given a randomly initialized dense network $W_{init}$, a learnable score $s$ is trained while keeping the weight values frozen. These scores are designed to encapsulate the importance of each weight for

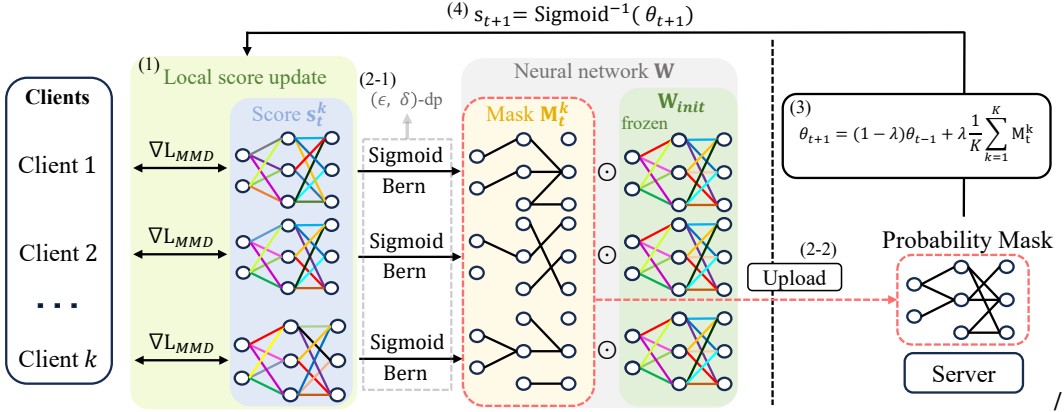

Figure 1: **Overview of PRISM.** PRISM finds the supermask for generative models in a FL scenario. At every round $t$, each client $k$ updates a local score $s_t^k$ via MMD loss (Step 1) and generates the privacy-preserving binary mask $M_t^k$ (Step 2-1), which is sent to the server. The server aggregates the masks to obtain the global probability $\theta_{t+1}$ (Step 3), which is converted to a score $s_{t+1}$ and broadcasted to the clients for the next round (Step 4). The global probability $\theta_{t+1}$ is gradually updated based on mask correlation $\lambda$ between $M_t^g$ and $M_{t-1}^g$.

the objective function. As the scores get iteratively updated, the EP algorithm progressively shrinks the model by applying binary masks to weights with higher scores, indicating their potentials to be included in the winning lottery ticket. The obtained SLT can be expressed as $W = W_{init} \odot M$, where $M$ is the obtained binary mask and $\odot$ denotes element-wise multiplication. SLT has been primarily explored within the context of classification tasks, while Yeo *et al.* (Yeo et al., 2023)] have recently shown that SLT can also be found in generative models.

## 3.2 DIFFERENTIAL PRIVACY

Sharing each client's model or gradient can potentially lead to a privacy risk. $(\epsilon, \delta)$-differential privacy (DP) (Dwork et al., 2006)], $(\alpha, \epsilon)$-Rényi-Differential privacy (RDP) (Mironov, 2017)] are commonly employed when tackling the privacy concerns in FL.

**Definition 1 (($\epsilon, \delta$)-Differential Privacy (Dwork et al., 2006)])** *A randomized mechanism* $\mathcal{M} : \mathcal{X} \to \mathcal{R}$ *is* $(\epsilon, \delta)$*-differential privacy, if for any two adjacent datasets* $\mathcal{D}, \mathcal{D}'$ *and for any measurable sets* $\mathcal{S}$: $Pr[\mathcal{M}(D) \in \mathcal{S}] \leq e^\epsilon Pr[\mathcal{M}(D') \in \mathcal{S}] + \delta$.

The above definition is designed to limit the impact of individual data points by introducing randomness into $\mathcal{M}$. The Gaussian mechanism (Mironov et al., 2019)] offers differential privacy guarantees by injecting Gaussian noise $\mathcal{N}(0, \sigma^2 I)$ to $\mathcal{M}$, where $\sigma^2 = \frac{2ln(1.25/\delta)\Delta_2^2}{\epsilon^2}$ and $\Delta_2^2$ is L2 sensitivity.

## 4 METHOD

We consider a FL setup with $K$ clients, where each client $k$ has its own local dataset $\mathcal{D}^k$. Starting from a randomly initialized model $W_{init}$, the clients aim to collaboratively obtain a global generative model $W^*$ that well-reflects all data samples in the system, i.e., in $\cup_{k=1}^{K} \mathcal{D}^k$.

**Overview of our approach.** Figure 1 shows the overview of our PRISM. PRISM finds a subnetwork with strong generative performance from the randomly initialized generative model $W_{init}$. Rather than updating $W_{init}$, its focus is to find an optimal binary mask $M^*$ that has either 1 or 0 in its element and construct the final global model as $W^* = W_{init} \odot M^*$. At a high-level, each client $k$ generates a binary mask $M_t^k$ based on its local dataset at every communication round $t$, which is aggregated at the server. After repeating the process for multiple rounds $t = 1, 2, \ldots, T$, PRISM produces the final supermask $M^* = M_T$. Our approach, which aims to find the SLT in a federated generative model setting, is fundamentally different from existing FedGAN methods. In the following, we describe the detailed training procedure of PRISM along with its advantages.

## 4.1 PRISM : PRIVACY-PRESERVING IMPROVED STOCHASTIC MASKING

**Local score updates with MMD loss.** Before training starts, the server randomly initializes the model $W_{init}$ and broadcasts it to all clients, which remains fixed throughout the training process. At the start of each round $t$, all clients download the score vector $s_t$ from the server, representing the importance of each parameter in $W_{init}$. Intuitively, if the score value of a specific parameter is high, the corresponding weight is more likely to be included in the final SLT. PRISM allows each client $k$ to update the score vector $s_t$ based on its local dataset to obtain $s_t^k$, which is used to generate the local mask. In this local score update procedure, we leverage maximum mean discrepancy (MMD) loss (Gretton et al., 2006; 2012)], providing stable convergence for training generative models (Li et al., 2017a;b; Bińkowski et al., 2018; Santos et al., 2019; Ramanujan et al., 2020; Yeo et al., 2023)]. The MMD loss measures the distance between two distributions by comparing their respective mean embeddings in a reproducing kernel hilbert space (RKHS) (Gretton et al., 2006; 2012)]. As in (Santos et al., 2019; Ramanujan et al., 2020)], we take VGGNet pretrained on ImageNet as a powerful characteristic kernel. Specifically, given the local dataset $\mathcal{D}^k = \{x_i^k\}_{i=1}^N$ of client $k$ and the fake image set $\mathcal{D}_{fake}^k = \{y_i^k\}_{i=1}^M$ produced by its own generator, the local objective function at each client $k$ is written as follows:

$$\mathcal{L}_{MMD}^k = \left\| \mathbb{E}_{x \sim \mathcal{D}^k}[\psi(x)] - \mathbb{E}_{y \sim \mathcal{D}_{fake}^k}[\psi(y)] \right\|^2 + \left\| \text{Cov}(\psi(\mathcal{D}^k)) - \text{Cov}(\psi(\mathcal{D}_{fake}^k)) \right\|^2, \quad (1)$$

where $\psi(\cdot)$ is a function that maps each sample to the VGG embedding space. Each client aims to match the mean and covariance between real and fake samples after mapping them to the VGG embedding space using kernel $\psi(\cdot)$. Based on Eq. 1, each client locally updates the scores to minimize the MMD loss according to $s_t^k = s_t - \eta \nabla \mathcal{L}_{MMD}^k$. Here, we note that the VGG network is utilized only for computing the MMD loss and is discarded when training is finished.

**Binary mask generation and aggregation.** After the local score update process, each client $k$ maps the score $s_t^k$ to a probability value $\theta_t^k \in [0, 1]$ as $\theta_t^k = Sigmoid(s_t^k)$, where $Sigmoid(\cdot)$ is the sigmoid function. The obtained $\theta_t^k$ is then used as the parameter of the Bernoulli distribution to generate the stochastic binary mask $M_t^k$, according to $M_t^k \sim Bern(\theta_t^k)$. Each client $k$ uploads only this binary mask $M_t^k$ to the server, significantly reducing the communication overhead. At the server side, the received masks are aggregated to estimate the global Bernoulli parameter as $\theta_{t+1} = \frac{1}{K} \sum_{k=1}^K M_t^k$, which can be interpreted as the probabilistic score reflecting the importance of the overall client's weights. $\theta_{t+1}$ is then converted to the score through the inverse of the sigmoid function according to $s_{t+1} = Sigmoid^{-1}(\theta_{t+1})$, which is broadcasted to the clients at the beginning of the next round. We provide the training details in Appendix C.

**Model initialization for storage efficiency.** When training is finished after $T$ rounds of FL, $\theta_T$ is obtained at the server. The supermask is then generated following $M^* \sim Bern(\theta_T)$, which is used to obtain the final global model as $W^* = W_{init} \odot M^*$. This final model $W^*$ can be stored efficiently even in resource-constrained edge devices, thanks to the model initialization strategy. When initializing $W_{init}$ in PRISM, we employ the standard deviation of Kaiming Normal distribution (He et al., 2015)], where the weight value in layer $l$ is sampled from the set $\sigma_l \sim \{-\sqrt{2/n_{l-1}}, \sqrt{2/n_{l-1}}\}$. Hence, by storing the scaling factor $\sqrt{2/n_{n_{l-1}}}$, each parameter in the initial model $W_{init}$ is already quantized to a 1-bit value (See Appendix E). This makes the final model exceptionally lightweight without extra pruning of quantization, which will be also showed via comparison in Section 5.5.

## 4.2 PRIVACY

To consider the situation of potential privacy treats, we incorporate $(\epsilon, \delta)$-differential privacy (DP) (Dwork et al., 2006)] into our framework. We provide a more detailed description related to the privacy preservation of PRISM and a specific scenario where PRISM can achieve additional privacy benefits in Appendix A and Appendix B, respectively.

## 4.3 MASK-AWARE DYNAMIC MOVING AVERAGE AGGREGATION

Data heterogeneity and privacy preservation pose significant challenges in accurately estimating the correct update direction. Both the previous and current global models contain valuable information about whether the local models are diverging Praneeth Karimireddy et al. (2019); Li et al. (2020);

Mendieta et al. (2022). To address this, we propose a mask-aware dynamic moving average aggregation (MADA) that leverages information from previous aggregation rounds in a mask-aware manner. To the best of our knowledge, MADA is proposed for the first time, introducing the mask-aware similarity to determine the moving average ratio on the server side. Specifically, when a client mask $M_t^k$ deviates significantly from the global model, the newly updated global mask $M_t^g$ will also differ considerably from the previous round's global mask $M_{t-1}^g$. However, such updates tend to favor the dominant clients. To mitigate this bias, the server calculates the *mask correlation* $\lambda$, which measures the similarity between the current and previous global masks ($M_t^g$ and $M_{t-1}^g$, respectively). The server then interpolates between the two global masks using $\lambda$ to adjust how much of the current mask should be incorporated. While various metrics can be used to compute the distance between masks, we employ the Hamming distance. We observe that using alternative distance metrics like cosine similarity yields similar performance (See Appendix L). The server-side aggregation process is defined as follows:

$$\theta_{t+1} = (1 - \lambda)\theta_{t-1} + \lambda \frac{1}{K} \sum_{k=1}^{K} M_t^k, \quad \lambda := dist(M_{t-1}^g, M_t^g), \tag{2}$$

This prevents excessive deviation by interpolating the aggregated mask with the previous Bernoulli parameter. As the global rounds progress, $\lambda$ gradually decreases, promoting stable convergence.

### 4.4 TRADING-OFF BETWEEN PERFORMANCE AND COMMUNICATION COST

While PRISM optimizes for minimal communication overhead and delivers satisfactory performance, it may not always meet the demand for higher-quality image generation. To address this, we introduces a flexible solution, PRISM* (See Appendix F).

## 5 EXPERIMENTS

In this section, we validate the effectiveness of PRISM on MNIST, FMNIST, CelebA, and CIFAR10 datasets. The training set of each dataset is distributed across 10 clients following either IID or non-IID data distributions[1]. We provide the detailed partitioning strategies for IID and non-IID simulation in Appendix D. In addition, for a fair comparison, we set $(9.8, 10^{-5})$-DP for all methods.

**Baselines.** We compare our method with several previous approaches for federated generative models under both *privacy-preserving* (with DP) and *privacy-free* (without DP) scenarios: In the privacy-preserving scenario, we consider DP-FedAvgGAN (Augenstein et al., 2019)] and GS-WGAN (Chen et al., 2020)] while in the privacy-free case, we adopt MD-GAN (Hardy et al., 2019)] and Multi-FLGAN (Amalan et al., 2022)]. In the case of Multi-FLGAN, the number of sync servers increases the communication cost quadratically, so we consider a $2\times2$ multi generator and discriminator setup.

**Performance metrics.** We evaluate the generative performance of each scheme using the commonly adopted metrics, including Fréchet Inception Distance (FID) (Heusel et al., 2017)], Precision & Recall (Kynkäänniemi et al., 2019)], Density & Coverage (Naeem et al., 2020)]. We further demonstrate the efficiency of PRISM by comparing the uplink cost (MB) (See Appendix G) at each FL round, the storage (MB) for the final models, and FLOPS (See Appendix J).

### 5.1 IID CASE

In this subsection, we examine an IID scenario where the training set of each dataset is uniformly distributed among clients. We compare various evaluation metrics under $(\epsilon, \delta)$-DP guaranteed setting (Table 1). Notably, PRISM outperforms current GAN-based models by a large margin. Figure 2 reveals that existing privacy-preserving methods often produce distorted images, particularly evident in CelebA, while our method tends to generate high-quality results. The above results support that PRISM can effectively find a SLT, achieving performance gains despite charging 48% less communication costs per round. It is worth noting that PRISM can further reduce the cost by applying extra techniques such as universal coding. Additionally, when comparing PRISM to PRISM[†] (MADA removed), PRISM demonstrates a significant improvement in overall performance.

---

[1]See Appendix H for additional results where a large number of clients and partial of clients participate.

Table 1: **Quantitative comparison in IID scenario with a privacy budget** $(\epsilon, \delta) = (9.8, 10^{-5})$. We compare FID, P&R, D&C, communication cost, and storage. Communication cost is the number of bytes exchanged between clients and server. † indicates that MADA is removed.

| Method (comm.cost) | Metric | MNIST | FMNIST | CelebA | Storage |
|---|---|---|---|---|---|
| GS-WGAN (15MB) | FID↓ | 71.1016 | 119.2589 | 230.7874 | 15MB |
| | P&R ↑ | 0.0975 / 0.1505 | 0.3694 / 0.0015 | 0.7951 / 0.0 | |
| | D&C ↑ | 0.0257 / 0.0367 | 0.1264 / 0.0347 | 0.165 / 0.0021 | |
| DP-FedAvgGAN (14MB) | FID↓ | 111.0855 | 118.5067 | 221.34 | 14MB |
| | P&R ↑ | 0.2586 / 0.0047 | 0.5318 / 0.0163 | 0.1008 / 0.0 | |
| | D&C ↑ | 0.0803 / 0.0141 | 0.2028 / 0.0341 | 0.0211 / 0.0013 | |
| PRISM† (5.75MB) | FID↓ | 48.5636 | 54.722 | 57.0573 | 7.25MB |
| | P&R ↑ | 0.3343 / 0.4265 | 0.5836 / 0.1574 | 0.4998 / **0.1221** | |
| | D&C ↑ | 0.1211 / 0.1151 | 0.2156 / 0.2432 | 0.2572 / 0.2189 | |
| PRISM (5.75MB) | FID↓ | **27.3017** | **46.1652** | **48.9983** | 7.25MB |
| | P&R ↑ | **0.4377 / 0.5576** | **0.6355 / 0.211** | **0.6435** / 0.076 | |
| | D&C ↑ | **0.1738 / 0.1982** | **0.4002 / 0.2971** | **0.4089 / 0.2415** | |

| (a) GS-WGAN | (b) DP-FedAvgGAN | (c) PRISM† | (d) PRISM |

Figure 2: **Qualitative results in IID scenario with a privacy budget** $(\epsilon, \delta) = (9.8, 10^{-5})$. We compare generated images from the models in Table 1 on MNIST, FMNIST, and CelebA. † indicates that MADA is removed.

## 5.2 NON-IID CASE

We investigate a more practical yet challenging non-IID scenario, where clients exhibit diverse local data distributions, posing a significant challenge to train generative models. To establish such an environment, we split the entire trainset based on shards-partitioning strategy[2]. Table 2 presents a quantitative comparison of baselines and our methods when DP is guaranteed. We observe that PRISM achieves SOTA performance under the non-IID scenario, even with complex CelebA datasets[3]. Fig-

---

[2]We also provide the results of Dirichlet-partitioned Non-IID in Appendix D.

[3]Since existing works on the field of federated generative models struggles with MNIST-level generation, we refer CelebA and CIFAR10 as complex dataset.

Table 2: **Quantitative comparison in non-IID scenario with** $(\epsilon, \delta) = (9.8, 10^{-5})$. We compare FID, P&R, D&C, communication cost, and storage. Communication cost refers to the number of bytes exchanged between clients and server. † indicates that MADA is removed.

| Method (comm.cost) | Metric | MNIST | FMNIST | CelebA | Storage |
|---|---|---|---|---|---|
| GS-WGAN (15MB) | FID ↓ | 338.6659 | 131.6166 | 228.9705 | 15MB |
| | P&R ↑ | 0.0 / 0.0 | 0.4186 / 0.0001 | 0.1363 / 0.0 | |
| | D&C ↑ | 0.0 / 0.0 | 0.1569 / 0.0297 | 0.0307 / 0.0025 | |
| DP-FedAvgGAN (14MB) | FID ↓ | 153.9325 | 146.632 | 222.8257 | 14MB |
| | P&R ↑ | 0.4371 / 0.0336 | **0.7207** / 0.0043 | 0.2331 / 0.0004 | |
| | D&C ↑ | 0.1049 / 0.004 | 0.2589 / 0.0164 | 0.0668 / 0.0016 | |
| PRISM† (5.75MB) | FID ↓ | 49.6273 | 83.0481 | 59.4877 | 7.25MB |
| | P&R ↑ | 0.3283 / 0.3844 | 0.4513 / 0.0775 | 0.4789 / **0.0898** | |
| | D&C ↑ | 0.1101 / 0.1022 | 0.2355 / 0.1428 | 0.2392 / 0.2058 | |
| PRISM (5.75MB) | FID ↓ | **34.2038** | **67.1648** | **39.7997** | 7.25MB |
| | P&R ↑ | **0.4386 / 0.4236** | 0.4967 / **0.1231** | **0.6294** / 0.0713 | |
| | D&C ↑ | **0.1734 / 0.1597** | **0.2748 / 0.1681** | **0.4565 / 0.2967** | |

(a) GS-WGAN    (b) DP-FedAvgGAN    (c) PRISM†    (d) PRISM

Figure 3: **Qualitative results in Non-IID scenario with a privacy budget** $(\epsilon, \delta) = (9.8, 10^{-5})$. We compare generated images from the models in Table 2 on MNIST, FMNIST, and CelebA. † indicates that MADA is removed.

ure 3 illustrates that despite the heterogeneity of the data, our methods successfully generate high quality images, while traditional methods exhibit subpar quality. Additionally, Figure 4 shows the overview of FID scores, final model parameters, and communication costs for each method.

## 5.3 PERFORMANCE WITHOUT DIFFERENTIAL PRIVACY

We further explore the performance of PRISM without applying differential privacy. Quantitative and qualitative comparisons with MD-GAN (Hardy et al., 2019)] and Multi-FLGAN (Amalan et al., 2022)], the current state-of-the-art under this condition, are shown in Table 3 and Figure 5. PRISM not only matches but also occasionally surpasses the performance of MD-GAN and Multi-FLGAN, all while significantly reducing communication overhead. Again, the fact that PRISM outperforms PRISM† clearly demonstrates the effectiveness of MADA. Here, Multi-FLGAN takes 10 times more

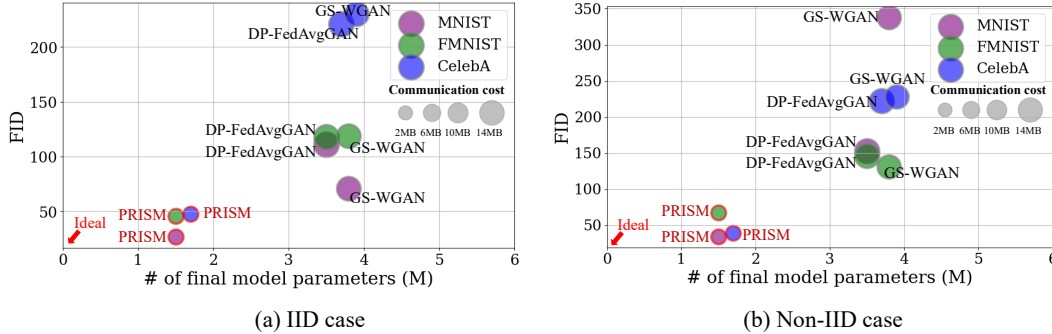

(a) IID case                                        (b) Non-IID case

Figure 4: **The performance of baselines and our PRISM with privacy budget** $(\epsilon, \delta) = (9.8, 10^{-5})$. X-axis represents the number of parameters of final generator, while Y-axis represents FID. The diameter of each circle denotes the required communication cost at every round. The ideal case is the bottom-left corner.

Table 3: **Quantitative comparison in non-IID scenario without DP.** We compare FID, P&R, D&C. Communication cost refers to the number of bytes exchanged between clients and server. † indicates that MADA is removed. We set $\alpha = 80$ for PRISM*.

| Method (comm.cost) | Metric | MNIST | FMNIST | CelebA | CIFAR10 | Storage |
|---|---|---|---|---|---|---|
| MD-GAN (14MB) | FID ↓ | 37.7971 | 55.5094 | 18.907 | 52.7159 | 14MB |
| | P&R ↑ | 0.3366 / 0.5435 | 0.5635 / 0.05 | 0.7612 / **0.6425** | **0.827** / 0.1968 | |
| | D&C ↑ | 0.1192 / 0.1405 | 0.3145 / 0.2033 | 0.7238 / 0.4267 | **1.2201** / 0.3829 | |
| Multi-FLGAN (52MB) | FID ↓ | 32.1014 | 125.9276 | 314.8386 | 163.0540 | 14MB |
| | P&R ↑ | 0.5659 / 0.3353 | 0.4781 / 0.0035 | 0.0 / 0.0 | 0.9345 / 0.0 | |
| | D&C ↑ | 0.3171 / 0.2709 | 0.24 / 0.0595 | 0.0 / 0.0 | 0.3638 / 0.0668 | |
| PRISM† (5.75MB) | FID ↓ | 15.2329 | 35.1448 | 24.2591 | 68.4238 | 7.25MB |
| | P&R ↑ | 0.7128 / 0.5289 | 0.7239 / 0.1049 | **0.7988** / 0.1868 | 0.65 / 0.1732 | |
| | D&C ↑ | 0.5106 / 0.4851 | 0.645 / 0.3768 | **1.0746** / 0.5809 | 0.5575 / 0.3031 | |
| PRISM (5.75MB) | FID ↓ | 9.698 | 32.7517 | 21.8567 | 61.1198 | 7.25MB |
| | P&R ↑ | 0.7665 / 0.6253 | **0.7614** / 0.1281 | 0.7835 / 0.1615 | 0.5924 / 0.2323 | |
| | D&C ↑ | **0.6088** / 0.6003 | **0.8361** / **0.4383** | 1.047 / 0.6079 | 0.4334 / 0.3171 | |
| PRISM* (15MB) | FID ↓ | **6.9568** | **29.0081** | **13.0209** | **35.5326** | 7.25MB |
| | P&R ↑ | **0.7717** / **0.7992** | 0.697 / **0.1572** | 0.7893 / 0.392 | 0.6662 / **0.3642** | |
| | D&C ↑ | 0.6082 / **0.6499** | 0.7002 / 0.4056 | 1.072 / **0.7396** | 0.5764 / **0.4481** | |

communication cost than PRISM due to multi GAN strategy. However, in FL setups, users may want to trade-off between the communication cost and generative performance. As shown in Table 3 and Figure 5, PRISM* outperforms across all benchmarks, including CIFAR10, while maintaining a communication cost comparable to MD-GAN and far lower than Multi-FLGAN. Further details on PRISM* can be found in Appendix F.

## 5.4 EFFECT OF DYNAMIC MOVING AVERAGE AGGREGATION

In this section, we empirically demonstrate the effectiveness of dynamic moving average aggregation using the MNIST dataset. Figure 6 visualizes local model updates over communication rounds $t$. At each global round $t$, clients receive the aggregated global mask $M_{t-1}^g$ and continue local training for several epochs. Afterward, the trained local mask $M_t^k$ is uploaded for the next communication round. We introduce the *local divergence* metric $\Delta_t := hd(M_t^g, M_t^k)$, which is defined as the Hamming distance between the received mask and the trained local mask to track the discrepancy of local model updates. For simplicity, we visualize the results for the first client, but similar trends were observed across other clients. In Figure 6, PRISM exhibits impressive FID scores and reduced local updates than PRISM†, clearly demonstrating that MADA not only restricts the local model divergence but also achieves significant performance gain across various challenging FL settings. This is accomplished by automatically obtaining $\lambda$ based on the current mask, without requiring additional regularization terms or hyperparameter tuning[4].

---

[4]The convergence of $\lambda$ in MADA and its relationship with client drift are discussed in Appendix I.

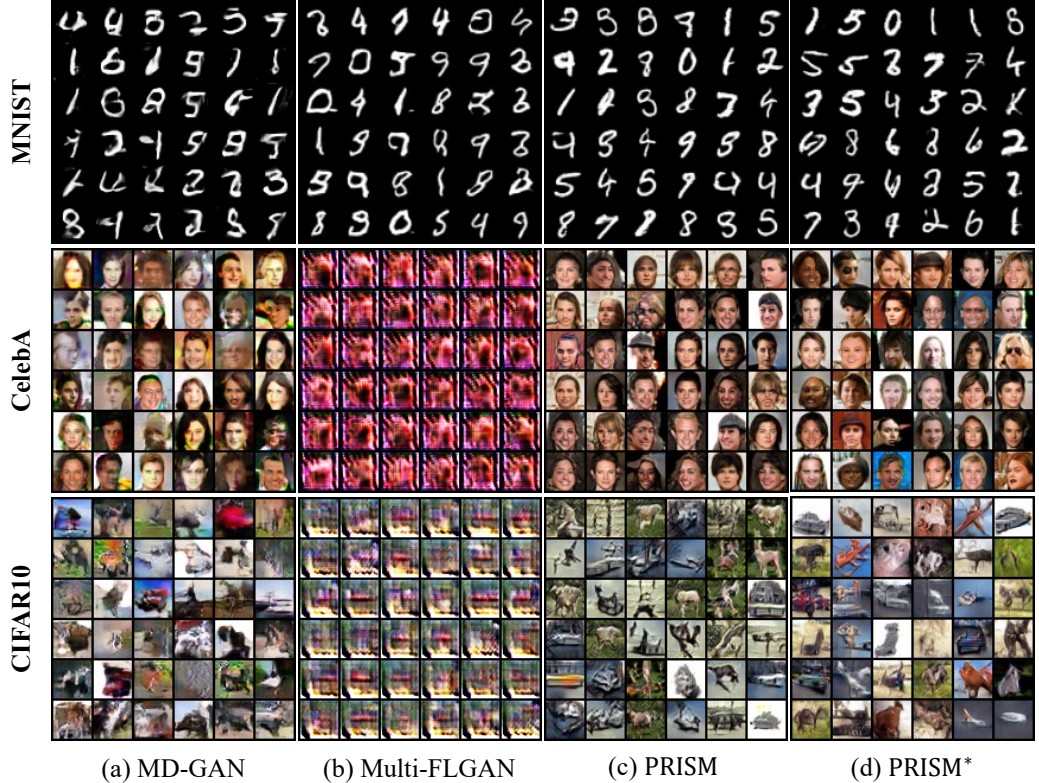

|               | (a) MD-GAN | (b) Multi-FLGAN | (c) PRISM | (d) PRISM* |

Figure 5: **Qualitative results in non-IID scenario without considering privacy budget.** Generated images from the models in Table 3 on MNIST, FMNIST, CelebA, and CIFAR10. Here, we set $\alpha = 80$ for PRISM*.

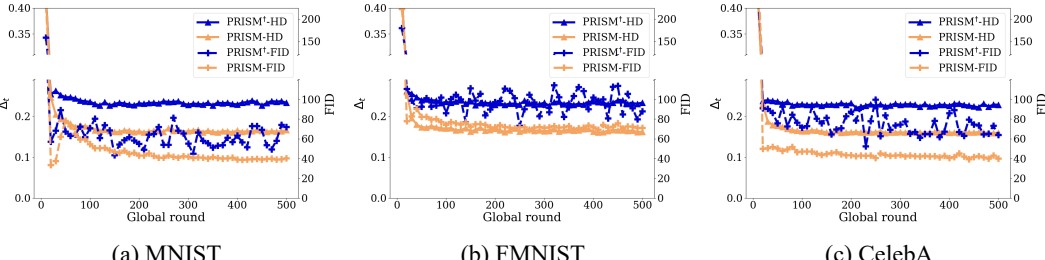

|            |            |            |
| (a) MNIST  | (b) FMNIST | (c) CelebA |

Figure 6: *Local divergence* $\Delta_t$ *and FID values in the non-IID and DP-considering scenario.*

## 5.5 RESOURCE EFFICIENCY AT INFERENCE TIME

The final model sizes of the baselines and our method are reported in Table 1, Table 2, and Table 3. Note that in addition to our method, applying various lossless compression techniques (*e.g.,* arithmetic coding (Rissanen & Langdon, 1979)]) can further reduce the required resources of PRISM.

## 6 CONCLUSION

While image generation has emerged as promising area in deep learning, the fusion of FL with generative models remains relatively unexplored. In this paper, we proposed PRISM, an efficient and stable federated generative framework that capitalizes on stochastic binary masks and MMD loss. To further enhance stability under non-IID and privacy-preserving scenario, we introduced a mask-aware dynamic moving average aggregation strategy (MADA) that mitigates client drift. Additionally, PRISM offers a hybrid mask/score aggregation method, allowing for a flexible and controllable trade-off between performance and efficiency. Our extensive experiments, including scenarios involving differential privacy and non-IID setups, demonstrated that PRISM is robust in unstable environments. To the best of our knowledge, PRISM is the first framework to consistently generate high-quality images with significantly reduced communication overhead in FL settings.

ACKNOWLEDGMENTS

This work was supported by the National Research Foundation of Korea (NRF) grant funded by the Korea government (MSIT) (No.2022R1C1C1008496) and Institute of Information & communications Technology Planning & Evaluation (IITP) grant funded by the Korea government (MSIT): (No.2022-0-00959, No.RS-2022-II220959 (Part 2) Few-Shot Learning of Causal Inference in Vision and Language for Decision Making), (No.RS-2022-II220264, Comprehensive Video Understanding and Generation with Knowledge-based Deep Logic Neural Network), (No.RS-2020-II201336, Artificial Intelligence Graduate School Program (UNIST)), (No.RS-2021-II212068, Artificial Intelligence Innovation Hub).

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

## A APPENDIX

## A PRIVACY

Diffenrential privacy (DP) and Rényi Differential Privacy (RDP) (Dwork et al., 2006; Mironov, 2017)] are the most popular definitions to analysis the privacy in FL environments. These help mitigate privacy concerns by limiting the contribution of individual data points. $(\epsilon, \delta)$-DP and $(\alpha, \epsilon)$-RDP basically calculates the distance of outcome for the algorithm of adjacent datasets.

**Definition 2 (($\epsilon, \delta$)-Differential Privacy)** *A randomized mechanism* $\mathcal{M} : \mathcal{X} \to \mathcal{R}$ *is* $(\epsilon, \delta)$-*differential privacy, if for any two adjacent datasets* $\mathcal{D}, \mathcal{D}'$ *and for any measurable sets* $\mathcal{S}$:

$$\boldsymbol{Pr}[\mathcal{M}(D) \in \mathcal{S}] \leq e^{\epsilon}\boldsymbol{Pr}[\mathcal{M}(D') \in \mathcal{S}] + \delta \quad (3)$$

**Definition 3 (($\alpha, \epsilon$) Rényi Differential Privacy)** *For two probability distributions P and Q, the Rényi divergence of order $\alpha > 1$ defined as follows:*

$$R_\alpha(P||Q) \triangleq \frac{1}{\alpha - 1} \log \mathbb{E}_{x \sim Q} (\frac{P(x)}{Q(x)})^\alpha \tag{4}$$

*then, a randomized mechanism $\mathcal{M} : \mathcal{X} \to \mathcal{R}$ is ($\alpha, \epsilon$) Rényi differential privacy, if for any two adjacent datasets $\mathcal{D}$, $\mathcal{D}'$ and for any measurable sets $\mathcal{S}$:*

$$R_\alpha(\mathcal{M}(D)||\mathcal{M}(D')) \leq \epsilon \tag{5}$$

**Theorem 1** *(Mironov, 2017)] showed that if $\mathcal{M}$ is ($\alpha, \epsilon$)-RDP guarantee, is also ($\epsilon + \frac{\log 1/\delta}{\alpha - 1}$)-DP.*

In this section, we provide more detailed explanation of privacy preserving in PRISM and also present updated results when DP is applied. To satisfy the ($\epsilon, \delta$)-DP, our goal is privatize the probability vector $\theta \in [0, 1]^d$ by adding gaussian noise $\mathcal{N}(0, \sigma^2)$, where $\sigma^2 = \frac{2ln(1.25/\delta)\Delta_2^2}{\epsilon^2}$ and $\Delta_2 = \max_{D,D'} ||\mathcal{M}(D) - \mathcal{M}(D')||_2$. When the local training is end, each client has scores $s \in \mathbb{R}^d$ to choose which weight to prune. Recall that probability $\theta \in [0, 1]^d$ can be obtained through sigmoid function. We inject gaussian noise and clip to $\tilde{\theta} \in [c, 1 - c]^d$, where c is a small value $0 < c < \frac{1}{2}$. In out setup, we fix it at 0.1. Now, we ensure $\tilde{\theta}$ is ($\epsilon, \delta$)-DP. For a fair comparison, we use ($\epsilon, \delta$) = ($9.8, 10^{-5}$) to PRISM and our baselines in all of our experiments. In addition, we regulate the global round to ensure that the overall privacy budget does not exceed $\epsilon$. To track the overall privacy budget, we employ subsampled moments accountant (Wang et al., 2019)]. We refer to the Opacus library which is the user-friendly pytorch framework for differential privacy (Yousefpour et al., 2021)].

(Imola & Chaudhuri, 2021; Isik et al., 2022)] have shown that performing post processing to already privatized vector $\tilde{\theta}$ such as Bernoulli sampling enjoys privacy amplification under some conditions. By doing so, the overall privacy budget becomes smaller $\epsilon_{amp} \leq \min\{\epsilon, d\gamma_\alpha(c)\}$, where $\gamma_\alpha(\cdot)$ is the binary symmetry Rényi divergence as expressed below:

$$\gamma_\alpha(c) = \frac{1}{\alpha - 1} \log(c^\alpha(1 - c)^\alpha + (1 - c)^\alpha c^{1-\alpha}), \tag{6}$$

where $\alpha$ refers to the order of the divergence. Note that $d$ limits the privacy amplification when the model size becomes large. Since PRISM assumes that the model size is large enough due to SLT, we focus on communication efficiency rather than privacy amplification.

## B  PRIVACY AMPLIFIED SCENARIO

In this section, we discuss about potential privacy benefit of PRISM under some conditions. Typical FL setting, a malicious third party can estimate $W_t^k - W_h^g$ from the communicated gradients. By analyzing these gradients, an attacker can extract information about the local data. However, with PRISM, only the binary masks $M_t^k$ and $M_t^g$ are exchanged, which hinders an attacker's efforts since they do not have access to the synchronized initial weight $W_{init}$ shared between the client and server.

## C  TRAINING DETAILS

In this section, we provide the detailed description of our implementations and experimental settings. In Table 4, we provide the model architectures used in our experiments. We use ResNet-based generator and set the local epoch to 100 and learning rate to 0.1. In addition, we do not employ training schedulers or learning rate decay. Our code is based on (Santos et al., 2019; Yeo et al., 2023)]. They employ the ImageNet-pretrained VGG19 network for feature matching by minimizing the Eq. 1. However, calculating the first and second moments require the large batch size to obtain the accurate statistics. To address this issue, (Santos et al., 2019)] introduces Adam moving average (AMA). With a rate $\lambda$, the update of AMA $m$ is expressed as follows:

$$m \leftarrow m - \lambda\text{ADAM}(m - \Delta), \tag{7}$$

Table 4: Generator architecture used in our experiments.

| Layer | Type | Input Channels | Output Channels | Kernel Size |
|-------|------|----------------|-----------------|-------------|
| FirstConv | Conv | 128 | 512 | (4, 4) |
| Resblock0 | Conv1 | 512 | 256 | (3, 3) |
| | Conv2 | 256 | 256 | (3, 3) |
| | BatchNorm2d | 512 | 512 | |
| | ReLU | - | - | - |
| | Bypass Conv | 512 | 256 | (1, 1) |
| Resblock1 | Upsample | - | - | - |
| | Conv1 | 512 | 256 | (3, 3) |
| | Bypass Conv | 512 | 256 | (1, 1) |
| | Upsample | - | - | - |
| | Conv2 | 256 | 128 | (3, 3) |
| | Conv3 | 128 | 128 | (3, 3) |
| Resblock2 | Conv1 | 128 | 64 | (3, 3) |
| | Conv2 | 64 | 64 | (3, 3) |
| | BatchNorm2d | 64 | 64 | - |
| | ReLU | - | - | - |
| LastConv | Conv | 64 | 1 | (3, 3) |
| | Tanh | - | - | - |

where ADAM denotes Adam optimizer (Kingma & Ba, 2014)] and $\Delta$ is the discrepancy of the means of the extracted features. Note that $\text{ADAM}(m - \Delta)$ can be interpreted as gradient descent by minimizing the L2 loss:

$$\min_m \frac{1}{2} \|m - \Delta\|^2. \tag{8}$$

This means the difference of statistics $(m - \Delta)$ is passed through a single MLP layer and updated using the Adam optimizer to the direction of minimizing Eq. 8. By utilizing AMA, Eq. 1 is formulated as $\mathcal{L}_{MMD}^k = \left\| \mathbb{E}_{x \sim \mathcal{D}^k}[\psi(x)] - \mathbb{E}_{y \sim \mathcal{D}_{fake}^k}[\psi(y)] \right\|^2 + \left\| \text{Cov}(\psi(\mathcal{D}^k)) - \text{Cov}(\psi(\mathcal{D}_{fake}^k)) \right\|^2$, Algorithm 1, 2 provides the pseudocode for MADA and PRISM* correspondingly. AMA is omitted to simply express the flow of our framework. See our code for pytorch implementation. We train the local generator for 100 local iterations with learning rate of 0.1. For the AMA layer, learning rate is set to 0.005. In addition, we use the Adam optimizer with $\beta_1 = 0.5, \beta_2 = 0.999$ to update the scores of the generators. After all clients complete their training, communication round is initiated. We set the global epoch to 150 for the MNIST dataset and 350 for the CelebA and CIFAR10 datasets. As we do not adjust the parameters, note that there is room for performance improvements through hyperparameter tuning.

## D   DATASET PARTITIONING STRATEGIES ACROSS CLIENTS

Here, we elaborate on the dataset partitioning strategies used to simulate the IID and non-IID setup. For non-IID setup, we used two strategies for partitioning datasets: 1) Shards-partitioned and 2) Dirichlet-partitioned.

**IID setup.** In the case of IID scenario, we assign equal-sized local datasets by uniformly sampling from the entire training set. This allows for a balanced distribution of data across the clients.

**Non-IID setup (Shards-partitioned).** In Section 5.2 and Section 5.3, we partition the MNIST, FMNIST, and CIFAR10 datasets into 40 segments based on the sorted class labels and randomly assign four segments to each client. For the CelebA dataset, which contains multiple attributes per image, defining a clear non-IID distribution for splitting is inherently ambiguous. In this work, we divide the dataset into tow partitions based on a pivotal attribute (gender, in our case). The total number of partitions corresponds to the number of clients, with each client being assigned either the positive or negative subset of images for the pivotal attribute. However, the remaining 39 attributes

---

**Algorithm 1** MADA

---

**Parameter:** learning rate $\eta$, communication rounds $T$, local iterations $I$
**Input:** local datasets $\cup_{k=1}^{K} \mathcal{D}^k$, ImageNet pretrained VGGNet $\psi$, random noise $z$

    **Server execute:**
    Initialize a random weight $W_{init}$ and score vector $s$, then broadcasts to all clients.
    **for** *round $t = 1, ..., T$* **do**
        **Client side:**
        **for** *each client $k \in [1, K]$* **do**
            $s_t^k = s_t$                                         ▷ Download score vector
            **for** *local iteration $i = 1, , , L$* **do**
                $\theta_t^k \leftarrow \text{Sigmoid}(s_t^k)$
                $M_t^k \sim \text{Bern}(\theta_t^k)$
                $W_t^k \leftarrow W_{init} \odot M_t^k$
                $\mathcal{D}_{fake}^k \leftarrow W_t^k(z)$                         ▷ Generate fake images
                Extract real and fake features $\psi(\mathcal{D}^k), \psi(\mathcal{D}_{fake}^k)$
                $s_t^k \leftarrow s_t^k - \eta \nabla \mathcal{L}_{MMD}^k(\psi(\mathcal{D}^k), \psi(\mathcal{D}_{fake}^k))$      ▷ Update local score vector
            **end for**
            $\bar{\theta_t^k} \leftarrow \text{Sigmoid}(s_t^k)$
            $\tilde{\theta_t^k} = \leftarrow \bar{\theta_t^k} + \mathcal{N}(0, I\sigma^2)$
            Clip to [c, 1-c]
            $M_t^k \sim \text{Bern}(\theta_t^k))$
            Upload binary mask $M_t^k$ to the server.
        **end for**
        **Server side:**
        $\hat{\theta}_{t+1} \leftarrow \sum_{k=1}^{K} M_t^k$                         ▷ Aggregate the received binary masks
        $s_{t+1} \leftarrow \text{Sigmoid}^{-1}(\hat{\theta}_{t+1})$
        $\lambda \leftarrow hd(M_t, \text{Bern}(\hat{\theta}_{t+1}))$                  ▷ Compute the hamming distance
        $s_{t+1} \leftarrow (1 - \lambda)s_t + \lambda s_{t+1}$
    **end for**
    Sample the supermask $M^* \sim \text{Bern}(\theta_T)$
    Obtain the final model $W^* \leftarrow W_{init} \odot M^*$

---

---

**Algorithm 2** PRISM$^*$

---

**Input:** ratio of score layer $\alpha$
**Output:** probability $\theta_t^k(100 - \alpha)$ and binary mask $M_t^k(\alpha)$

    **Client side:**
    **for** *layer $l = 1, ..., L$* **do**
        **if** IsScoreLayer(l,$\alpha$,L) **then**
            Return probability $\theta_t^k(l)$
        **else**
            Return binary mask $M_t^k(l)$
        **end if**
    **end for**

---

are still shared among clients, resulting in relatively weak heterogeneity. This explains why the performance drop on CelebA is not as significant compared to the IID case. For a more detailed discussion, please refer to Appendix D.

**Non-IID setup (Dirichlet-partitioned).** We further explore a more realistic and challenging non-IID scenario, where datasets are partitioned using Dirichlet distribution. Specifically, we set Dirichlet parameter $\alpha = 0.005$ to create a more label-skewed distribution. For CelebA, we assign data to clients such that each client possesses the pivotal attribute in different proportions using Dirichlet distribution. For example, client 1 has 60% male and 40% female, while client 2 has 20% male and

Table 5: **Quantitative comparison in Dirichlet non-IID scenario with** $(\epsilon, \delta) = (9.8, 10^{-5})$. We compare FID, P&R, D&C. † indicates that MADA is removed.

| Method | Metric | MNIST | FMNIST | CelebA |
|---|---|---|---|---|
| GS-WGAN | FID ↓ | 128.4401 | 134.4054 | 108.8792 |
| | P&R ↑ | 0.0851 / 0.0633 | 0.3927 / 0.0001 | **0.8891** / 0.0154 |
| | D&C ↑ | 0.0196 / 0.0071 | **0.1252** / 0.0348 | 0.2165 / 0.052 |
| DP-FedAvgGAN | FID ↓ | 175.3729 | 209.1346 | 323.3559 |
| | P&R ↑ | 0.0408 / **0.1982** | 0.0814 / **0.0421** | 0.1357 / 0.0 |
| | D&C ↑ | 0.0102 / 0.0048 | 0.0214 / 0.0123 | 0.0271 / 0.0001 |
| PRISM† | FID ↓ | 74.739 | 137.7545 | 57.4051 |
| | P&R ↑ | 0.2209 / 0.1156 | 0.3041 / 0.0274 | 0.482 / **0.1058** |
| | D&C ↑ | 0.0619 / 0.0415 | 0.1086 / 0.0312 | 0.2317 / 0.1934 |
| PRISM | FID ↓ | **70.9146** | **137.7359** | **38.0283** |
| | P&R ↑ | **0.4119** / 0.1563 | **0.3394** / 0.0378 | 0.6305 / 0.0746 |
| | D&C ↑ | **0.1513** / **0.0759** | 0.1139 / **0.0394** | **0.4493** / **0.3144** |

Table 6: **Quantitative comparison in Dirichlet non-IID scenario without DP.** We compare FID, P&R, D&C. † indicates that MADA is removed. We set $\alpha = 80$ for PRISM*.

| Method | Metric | MNIST | FMNIST | CelebA | CIFAR10 |
|---|---|---|---|---|---|
| MD-GAN | FID ↓ | 106.3468 | 86.0443 | 30.7787 | **54.38** |
| | P&R ↑ | **0.7109** / 0.2431 | 0.5586 / 0.0359 | 0.7612 / **0.6425** | **0.8941** / 0.1903 |
| | D&C ↑ | **0.6884** / **0.4895** | 0.2137 / 0.0917 | 0.7238 / 0.4267 | **1.4087** / **0.355** |
| Multi-FLGAN | FID ↓ | | | | |
| | P&R ↑ | | diverge | | |
| | D&C ↑ | | | | |
| PRISM† | FID ↓ | 43.3057 | 76.1413 | 21.0935 | 89.0257 |
| | P&R ↑ | 0.6186 / 0.216 | **0.6093** / 0.023 | **0.797** / 0.2229 | 0.6916 / 0.0854 |
| | D&C ↑ | 0.3219 / 0.1799 | **0.3443** / 0.1163 | 0.991 / 0.6533 | 0.5885 / 0.2157 |
| PRISM | FID ↓ | **31.6191** | 73.8701 | 20.0883 | 74.1502 |
| | P&R ↑ | 0.5871 / 0.36 | 0.5239 / 0.0378 | 0.782 / 0.2024 | 0.572 / 0.1601 |
| | D&C ↑ | 0.2828 / 0.2328 | 0.2919 / 0.1054 | **1.1148** / 0.6893 | 0.3838 / 0.2535 |
| PRISM* | FID ↓ | 38.8984 | **72.8571** | **13.6906** | 57.9337 |
| | P&R ↑ | 0.5671 / **0.5423** | 0.4926 / **0.045** | 0.784 / 0.3266 | 0.6395 / **0.2645** |
| | D&C ↑ | 0.27 / 0.2251 | 0.2628 / **0.1177** | 0.9627 / **0.7352** | 0.567 / 0.327 |

80% female. To the best of our knowledge, methods for splitting multi-label datasets into extreme non-IID scenarios have not been extensively explored. Table 5 and Table 6 present a quantitative comparison of baselines and our methods under both with and without DP. PRISM demonstrates robust performance even under highly heterogeneity distributions. Note that Multi-FLGAN fails to successfully train generative model under highly heterogeneity.

## E  EFFICIENT STORAGE AND QUANTIZATION STRATEGY IN PRISM

In this section, we provide further detail description regarding storage efficiency of PRISM. Aforementioned in Section 4.1, the final model can be expressed as $W^* = W_{init} \odot M^*$. Since $M_{fianl}$ is lightweight binary masks, we need to focus on the storage of $W_{init}$. Note that $W_{init}$ is initialized as signed constant using Kaiming Normal distribution. In other words, the initialized parameter of layer $l$ becomes $\{+\sqrt{2/n_{l-1}}, -\sqrt{2/n_{l-1}}, \cdots, +\sqrt{2/n_{l-1}}\} \in \mathbb{R}^d$. This design makes storing the scaling factor $v = \sqrt{2/n_{l-1}}$ be enough to represent each parameter, represented as $W_{init} = v \odot M_{sign}$, where $M_{sign}$ is defined as $\{+1, -1, \cdots, +1\} \in \mathbb{R}^d$. Our memory-efficient approach is closely related to ternary quantization (Zhu et al., 2016)], which typically compresses neural network weights into $\{-1, 0, 1\}$ through thresholding and projection. However, PRISM dis-

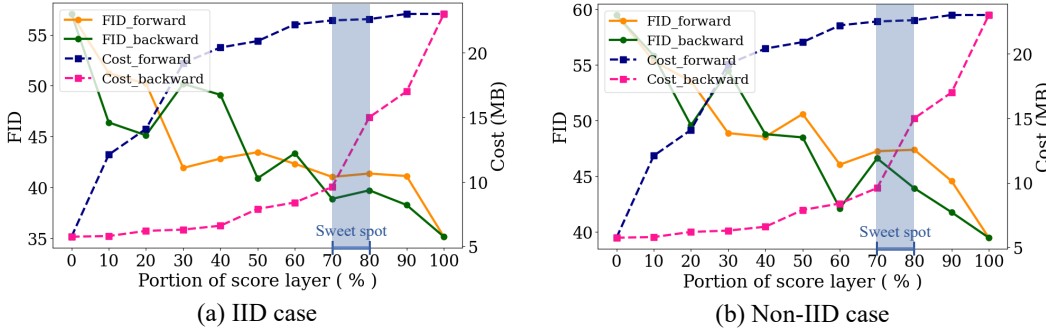

Figure 7: **Analysis of PRISM* using CIFAR10 dataset.** The *backward path* selects $\alpha\%$ of *score layers* from deeper layers, closer to the output, while the *forward path* chooses from the opposite end, nearer to the input. Solid-line demonstrates FID following each direction while dash-line shows communication cost (MB) of each path.

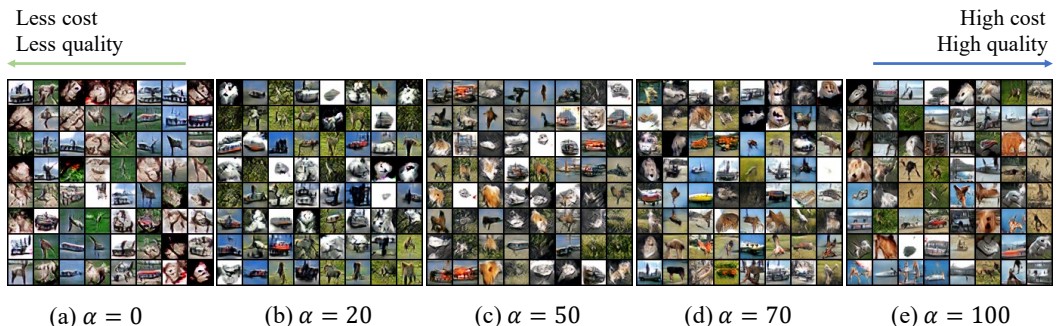

Figure 8: **The effect of adjusting $\alpha$ of PRISM*.** Qualitative comparison according to $\alpha$. Note that $\alpha = 0$ is identical to PRISM.

tinguishes itself by directly initialized frozen weights as signed constants using the Kaiming Normal distribution. Since storing a signed array is negligible in terms of memory overhead, this design makes the final model exceptionally lightweight without extra pruning or quantization.

## F  ANALYSIS OF HYBRID AGGREGATION

In this section, we further analyze PRISM*, which leverages both binary mask and score communications. To explore the trade-off between communication cost and generative capability, we consider two strategies: the *backward path* and *forward path*. The *backward path* progressively increases the number of *score layers* from deeper layers to earlier layers. Conversely, in the *forward path*, we select *score layers* from earlier layers to deeper layers. Figure 7 visually demonstrates the trade-off between communication cost and FID of both strategies. The FID gradually improves as we increase $\alpha$ values in both cases. Note that the additional communication cost of the *backward path* tends to increase more smoothly. Based on these observations, we adopt the *backward path* for Table 3 and Figure 5. Figure 8 provides a comprehensive comparison across a wide range of $\alpha$ values, showing that PRISM* consistently produces high quality images. Note that all experiments are conducted in privacy-free scenario.

## G  COMMUNICATION COST DURING DOWNLINK PROCESS

In a typical FL setup, the server possesses powerful computational capabilities, whereas the clients do not. As a result, several prior studies primarily focus on addressing the limited bandwidth of clients (Kim et al., 2024a; Hu et al., 2023; Yi et al., 2022)]. Accordingly, we have concentrated on the reducing communication cost during the uplink process, transmitting a float-type score during downlink to mitigate the information loss. Last but not least, binary mask communication can also

be employed in the downlink to further enhance communication efficiency, despite the occurrence of information loss during the conversion of the global mask into local scores.

Table 7: **Quantitative results on cross-device environment.** We set the number of clients to 50 and participant ratio to 20%.

| Case | Metric | MD-GAN | DP-FedAvgGAN | GS-WGAN || PRISM |
|---|---|---|---|---|---|
| Non-IID w/ DP | FID ↓ | N/A | 118.3975 | 98.6553 | **34.7157** |
| | P&R ↑ | N/A | 0.1095 / 0.3723 | **0.8477** / 0.0359 | 0.4344 / **0.3401** |
| | D&C ↑ | N/A | 0.0301 / 0.0289 | **0.2621** / 0.0105 | 0.1692 / **0.1476** |
| Non-IID w/o DP | FID ↓ | 15.4119 | N/A | N/A | **14.3168** |
| | P&R ↑ | 0.7305 / 0.359 | N/A | N/A / N/A | **0.7533 / 0.4757** |
| | D&C ↑ | 0.5266 / 0.3803 | N/A | N/A | **0.5804 / 0.5293** |

## H CROSS-DEVICE ENVIRONMENTS

PRISM is designed to provide efficient communication cost, making it ideal for cross-device environments with numerous clients and limited bandwidth. Typically, cross-device setting involves a large number of clients and an unreliable network, such as those encountered in iPhone and wearable devices. This complexity makes it difficult to ensure that all clients in every communication round. To simulate this cross-device environment, we set the number of clients to 50 and the participant ratio as 20%. Table 7 shows the quantitative results using MNIST dataset, both with and without DP. PRISM demonstrates a clear performance advantage over other baselines and is well-suited for the cross-device environment.

## I ANALYSIS OF CONVERGENCE OF $\lambda$ IN MADA

In this section, we provide a detailed discussion on how the reduction in $\lambda$, which measures the similarity between two consecutive global masks, helps mitigate client drift.

One possible explanation is that MADA implicitly modulates the learning rate for global updates on the server-side without requiring an explicit learning rate scheduler. This behavior reduces the need for client-side learning rate scheduling to regularize local objectives. To illustrate this, consider a local mask update represented as $w_k^t = w_k^{t-1} - \eta \cdot \Delta_k^t$, where $\eta$ is constant learning rate, and $\Delta_k^t$ is local model update from client $k$. In standard FedAvg, the global update can be expressed as $w_{\text{FedAvg}}^t = w_k^{t-1} - \eta \cdot \Sigma_k \Delta_k^t$, which aggregates all client updates equally. In contrast, MADA introduces a mask-aware aggregation mechanism, and the global model is formulated as $w_{\text{MAGA}}^t = (1 - \lambda)w_k^{t-1} + \lambda \cdot w_k^t$, where $w_k^t = w_k^{t-1} - \eta \cdot \Delta_k^t$. Substituting the local mask update into this equation, $w_{\text{MAGA}}^t = (1 - \lambda)w_k^{t-1} + \lambda \cdot (w_k^{t-1} - \eta \cdot \Sigma_k \Delta_k^t) = w_k^{t-1} - \eta \cdot \lambda \Sigma_k \Delta_k^t$, indicating that $\lambda$ effectively scales the global updates step size. When $\lambda$ is large, the global model closely follows the client updates, while a small $\lambda$ reduces the influence of client updates, acting as a dampening factor.

The intuition behind this behavior is rooted in the nature or early-stage training. During the initial rounds, the model is far from convergence, resulting in large gradient norms and substantial parameter changes. Consequently, consecutive global masks exhibit significant differences, leading to a large $\lambda$. As training progresses and the model stabilizes, mask updates become more similar, decreasing $\lambda$. This gradual reduction in $\lambda$ prevents excessive global updates, stabilizing convergence and mitigating client drift. We elaborate on the behavior of client drift. Even when client-drift is biased toward a specific dominant client, the update difference for the dominant client naturally decreases from the perspective of global support, whereas the difference from the local solutions of other clients increases. This pattern is reflected in the aggregated model, and the difference between consecutive global models can partially capture this behavior (e.g., indicated by a large $\lambda$). As training progresses and the model approaches the global optimum across local objectives, the differences between consecutive global models gradually decrease, leading to a reduction in $\lambda$. Conversely, in the absence of client drift, such as in IID scenarios, the model learns fairly across all clients and naturally converges without notable fluctuations in global model differences.

In summary, MADA updates the global model in a way that avoids local divergence, gradually reducing global model differences even without learning rate decay. As shown in Figure 6, $\lambda$ decreases sharply in the early stages of training due to significant local divergence but gradually converges over time. This result suggests that our understanding aligns with the behavior of $\lambda$.

Table 8: Comparison of FLOPS between PRISM and other baselines.

|  | DP-FedAvgGAN | GS-WGAN | PRISM |
|---|---|---|---|
| FLOPS | 0.002 B | 1.94 B | 0.34 B |

## J    COMPUTATION COST ANALYSIS

One might say that SLT process within PRISM increases computational complexity, particularly in large networks. As the number of model parameters grows, the required score parameters also scale linearly. However, in SLT, weight parameters are not optimized via gradient descent. The only additional computations stem from the sigmoid and Bernoulli processes introduced during the binary masking procedure. While these operations introduce some additional computational overhead, they remain negligible in the context of GPU-accelerated gradient descent. To demonstrate that the additional computational cost is not substantial, we compare floating-point-operations per second (FLOPS) in Table 8. GS-WGAN exhibits increased FLOPS due to spectral normalization, which is applied to stabilize GAN training. In contrast, PRISM, despite incorporating the SLT process, maintains a sufficiently low FLOPS, ensuring its computational efficiency.

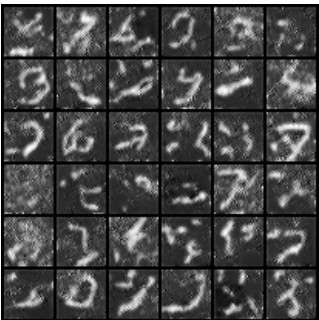

Figure 9: Generated images of PRISM where ddpm is used instead of MMD loss.

## K    SCALABILITY OF PRISM

Due to the challenges associated with training generative models in FL setups, research on federated generative models has primarily focused on simple architectures (*e.g.,* GANs) and datasets, such as MNIST and FMNIST level generation. However, it is important to note that the field of federated generative models remains far behind the current trend of using more complex and powerful models, such as diffusion models, and large-scale datasets. This gap arises from the inherent difficulties in adapting these advanced models to FL setups. Existing works have struggled to achieve stable performance in challenging setups, such as non-IID and privacy-preserving cases, even with MNIST-level datasets. Even for centralized differential privacy generative models (Dockhorn et al., 2022; Jiang et al., 2024)],training generative models is highly challenging and often unstable. Due to these obstacles, prior studies have primarily focused on simpler datasets, such as MNIST, and have been largely restricted to GANs.

### K.1    EXTEND TO DIFFUSION MODEL

We further investigate the applicability of PRISM to larger and mode powerful generative models, such as diffusion models. Specifically, we retain the core concept of identifying SLT but replaced

the MMD loss with DDPM loss (Ho et al., 2020)] to guide score updates. Notably, all other configurations remain unchanged. We conduct experiments on the MNIST dataset under a non-IID, privacy-free scenario. Figure 9 presents the generated images on the server-side, which exhibit slight noisiness. We anticipate that by improving the approach for identifying SLT within diffusion models and further curating the experimental setup, we can achieve more promising results. On the other hand, directly applying the MMD loss-based approach used in PRISM, which requires generating a set of samples for each update, poses practical challenges due to the well-known limitations of DDPM, including slow sampling and convergence rates. These limitations lead to significant computational costs, making the approach practically infeasible. Since our primary focus is lightweight communication cost and stable performance, we leave this future work, efficiently incorporating the powerful and diffusion model in federated settings.

## K.2 EVALUATION ON LARGE-SCALE DATASETS

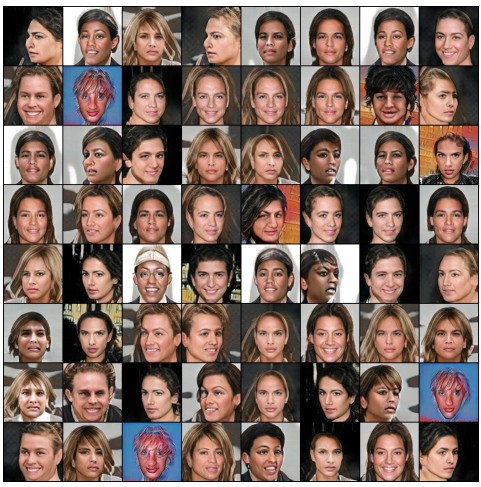 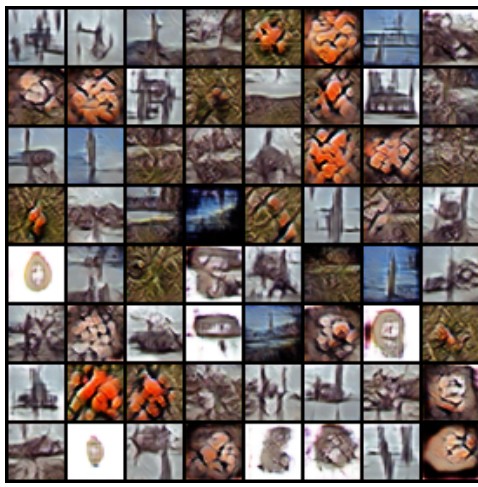

Figure 10: Generated images of CelebA 128x128 dataset and CIFAR100 dataset under non-IID and privacy-free scenario.

Table 9: Performance of PRISM using CelebA 128x128 and CIAFR100 datasets under non-IID and privacy-free scenario.

| Dataset | FID | P& R | D& C |
|---------|------|------|------|
| CelebA 128 | 40.2927 | 0.7738 / 0.006 | 1.0027 / 0.348 |
| CIFAR100 | 74.1609 | 0.6655 / 0.0719 | 0.602 / 0.3121 |

Note that the field of federated generative models have struggled to achieve stable performance in challenging setups, such as non-IID and privacy-preserving cases, even with MNIST-level datasets. However, even under these conditions, PRISM is the first to consistently demonstrate stable performance under these conditions. Despite the poor conditions of federated generative models, it is important to address whether PRISM can be extended to large-scale datasets under limited resources. We provide quantitative and qualitative results on large-scale datasets in Table 9 and Figure 10, such as CelebA 128x128 and CIFAR100 datasets, under non-IID and privacy-free scenario. Both experiments are conducted under exactly same settings as in Section 5. As a result, PRISM demonstrates acceptable performance on high-resolution images; however, it reveals certain limitations when applied to large-scale datasets with diverse classes, such as CIFAR100.

Table 10: FID comparison where distance metric is cosine similarity for MADA.

| Distance | Case | MNIST | FMNIST | CelebA |
|----------|------|-------|--------|--------|
| PRISM w/ hd | IID, DP | **27.3017** | 46.1652 | **48.9983** |
| PRISM w/ cos | | 27.7895 | **44.4084** | 49.0747 |
| PRISM w/ hd | Non-IID, DP | **34.2038** | **67.1648** | **39.799** |
| PRISM w/ cos | | 34.9577 | 69.2994 | 51.1734 |

## L  SELECTION OF OTHER DISTANCE METRIC

Table 10 provide the FID performance of PRISM when using cosine similarity instead of hamming distance for MADA. PRISM with cosine similarity achieves competitive performance compared to the scheme using Hamming distance in most cases. Although the performance degrades for CelebA in the non-IID scenario, it still outperforms all baseline methods, further demonstrating the robustness of the approach.

## M  THE EFFECT OF MASKING MECHANISM

Table 11: Effect of mask selection on MNIST.

| Dataset | Metric | Random | Top-k% | Bernoulli (ours) | Weight |
|---------|--------|--------|--------|------------------|--------|
| MNIST | FID $\downarrow$ | 391.8257 | 20.3616 | **12.7373** | 5.9895 |
| | P&R $\uparrow$ | 0.0 / 0.0006 | 0.5232 / 0.3782 | **0.7323 / 0.5904** | 0.6783 / 0.8414 |
| | D&C $\uparrow$ | 0.0 / 0.2511 | 0.2854 / 0.0 | **0.5556 / 0.5313** | 0.446 / 0.678 |

In Table 11, we compare the effect of the mask extraction algorithm on generative model training. For this, we relied on prior studies on the existence of SLT and the convergence of MMD with a characteristic kernel [46, 14]. Our results empirically show SLT's convergence under mask averaging generative FL setting. We also explore the Random and Top-k% algorithms. The Bernoulli method employed in PRISM outperformed top-k%, while Random and Weight represent the lower and upper bounds of performance achievable by SLT, respectively.

## N  EVALUATION ON THE CENTRALIZED SETTING

Table 12: Comparison of PRISM with centralized setting.

| Method | Metric | MNIST | FMNIST | CelebA |
|--------|--------|-------|--------|--------|
| PRISM | FID $\downarrow$ | 34.2038 | 67.1648 | 39.7997 |
| | P&R $\uparrow$ | 0.4386 / 0.4236 | 0.4967 / 0.1231 | 0.6294 / 0.0713 |
| | D&C $\uparrow$ | 0.1734 / 0.1597 | 0.2748 / 0.1681 | 0.4565 / 0.2967 |
| PRISM (vanilla) | FID $\downarrow$ | 5.8238 | 5.5004 | 19.1512 |
| | P&R $\uparrow$ | 0.6913 / 0.851 | 0.6985 / 0.8534 | 0.6621 / 0.3895 |
| | D&C $\uparrow$ | 0.4689 / 0.679 | 0.4864 / 0.6965 | 0.5348 / 0.5947 |

In Table 12, we report the results of models trained under FL vs. vanilla (centralized data) setups. PRISM shows performance degradation due to privacy, data heterogeneity, and communication overhead in FL settings. However, centralized training is not applicable in scenarios where distributed samples cannot be shared, which is the primary focus of our work. Therefore, our original submission did not include this comparison, consistent with existing FL research.

## O    ADDITIONAL RESULTS

### O.1    DIFFERENT RANDOM INITIALIZATION OR SEED

Table 13: Quantitative performance of PRISM across different random seeds on the MNIST dataset.

| IID, DP | seed | FID | P& R | D& C |
|---|---|---|---|---|
| PRISM | 30 | 27.3017 | 0.4377 / 0.5576 | 0.1738 / 0.1982 |
| | 123 | 26.8707 | 0.4342 / 0.5399 | 0.174 / 0.2047 |
| | 1234 | 26.9664 | 0.4417 / 0.5842 | 0.1785 / 0.1997 |
| **Non-IID, DP** | **seed** | **FID** | **P& R** | **D& C** |
| PRISM | 30 | 34.2048 | 0.4386 / 0.4236 | 0.1734 / 0.1597 |
| | 123 | 34.0586 | 0.4372 / 0.3306 | 0.1732 / 0.1618 |
| | 1234 | 32.9423 | 0.3921 / 0.4182 | 0.1496 / 0.1587 |

PRISM learns a binary mask over frozen, randomly initialized weights. One might argue that the way model parameters are initialized can significantly impact performance. To validate the robustness of PRISM, we present quantitative performance across various random seeds using the MNIST dataset in Table 13. Note that the random seed 30 in the first row of each table corresponds to the default configurations described in Section 5 and all of experimental setup is identical to Section 5.1 and Section 5.2.

### O.2    ABLATION STUDY ON THE LEARNING RATE FOR BASELINES

Table 14: Baseline's FID performance for adjusting learning rate using MNIST dataset.

| Non-IID, DP | 1e-4, 1e-4 (default) | 1e-3, 1e-4 | 1e-5, 1e-4 | 1e-4, 1e-3 | 1e-4, 1e-5 |
|---|---|---|---|---|---|
| GS-WGAN | diverge | 108.0657 | diverge | diverge | 96.1892 |
| **Non-IID, DP** | **1e-3, 5e-4 (default)** | **1e-3, 5e-3** | **1e-2, 5e-4** | **1e-4, 5e-4** | **1e-3, 1e-5** |
| Dp-FedAvgGAN | 153.9325 | 206.759 | diverge | 177.3752 | 232.2336 |

In this section, we present the additional experiments to validate the fairness of our entire experiments. In Table 14, we sweep learning rates for GS-WGAN and DP-FedAvgGAN and a FID performance on MNIST dataset with Non-IID and privacy-preserving scenario. Since they utilize GANs, there are separate learning rates for the generator and discriminator, and we examined these combinations. In the tables below, the top row represents the discriminator's lr / generator's lr, while the subsequent row reports the FID scores. Due to the notorious instability of GAN training caused by the coupled learning dynamics between the generator and discriminator, we observed divergence in some settings. For GS-WGAN, the best performance was achieved with 1e-4 / 1e-5, but it still failed to generate MNIST images properly.

## P    CONTEXTUALIZE THE PRACTICAL SCENARIOS

Federated learning (FL) is commonly motivated by real-world scenarios such as healthcare, finance, and mobile devices, where direct data sharing is infeasible due to privacy concerns. In this context, our work does not aim to generate private data directly but instead focuses on enabling stable and effective generative modeling under these constraints. For instance, in healthcare, PRISM allows multiple institutions to collaboratively train generative models without sharing sensitive patient data. The resulting models can facilitate privacy-preserving synthetic data generation, imputation of missing medical records, data augmentation for low-data scenarios, and solving inverse problems such as denoising and reconstruction. Similarly, in mobile devices, PRISM enables decentralized training of generative models while preserving user privacy, which can enhance personalized content generation and user-specific data augmentation.

