# OpenReview forum: "PRISM: Privacy-Preserving Improved Stochastic Masking for Federated Generative Models"
_ICLR.cc/2025/Conference — ICLR 2025 Poster_

### Official Review · Reviewer_EU7U · 2024-10-22

**Soundness:** 2
**Presentation:** 3
**Contribution:** 2
**Rating:** 6
**Confidence:** 3

**Summary:**

This paper uses the strong lottery ticket hypothesis to search for valuable weights within initializations. They optimize for utilitarian weight masking structures, which provides communication efficiency and memory benefits. They apply their approach to federated learning, specifically within the context of generative models. The authors argue that PRISM is robust for non-IID data and promotes resource efficiency.

**Strengths:**

(1) Figure 1 is quite effective in compactly communicating the main ideas.

(2) Originality and significance: This paper is an interesting read. I will reserve my opinion on these two factors until after further clarification from the authors for (1) in weaknesses.

(3) Clarity and quality: It is clear that much effort has been put into composing the related works.

**Weaknesses:**

(1) Can the authors help me understand the differences between PRISM, and the works [1-2]? The central methodology of this paper appears to just be a combination of the two papers, up to some minor changes.

(2) The MNIST digits generated in Figures 2/3 for PRISM and baselines look problematic. As these are tasks that we would expect even a simple VAE to be able to do, I am concerned about the significance of the contributions given in this paper. The fact that Figure 2 is in the IID scenario is also concerning, as one might expect contributable training paradigms to be able to generate MNIST digits properly. I am beginning to wonder if I am missing the point of the paper. Moreover, I do not agree with the assertion that CIFAR10 and CelebA are "complex datasets" which allows evaluation of "state-of-the-art image generation" in line 083. Figure 3 gives some sensible-looking digits. However, this is an empirical paper, without theoretical results. Additionally, the central idea of training for lottery subnetworks has already been examined in prior works, e.g., cited above. Given this, MNIST, CelebA, and CIFAR-10 are not difficult enough to yield publishable empirical results by themselves, and there are far more informative datasets out there (e.g., ImageNet subsets).

(3) In Appendix G, the authors state that they do not include comparisons with centralized settings as they study settings in which "distributed samples cannot be shared". I think the concern here is that the evaluated datasets are so mature, and generative central training on them easily practicable on even modern Laptops, such that if the evaluated methods cannot even beat the central model, it is difficult to see the contribution being made. Again, my assessment would differ significantly if this was a theoretical paper, but this paper is mainly empirical. Are there any reasonable, practical settings in which we would be forced to train in such restrictive scenarios? Training generative image models on iPhones with user data?

(4) Could the authors point me to where they discuss (or add in a discussion of):

(a) What is the architecture specification used in the paper?

(b) How are the hyperparameters tuned?

Regarding (a), it would be very helpful to explicitly discuss the architecture details in the text so that it is self-contained (c.f., Table 2 in [1]). That way, readers do not have to go line-by-line through the code link to get this information.

(5) The empirical setup and discussions seem to be mismatched. The prior discussions seem to focus on communication costs, memory, privacy, etc. This strongly suggests the resource-restricted cross-device setting in FL, which is especially the case given the simplicity of model architectures as well as datasets being used for generation. However, the experiments use 10 clients, which is clearly in the cross-silo setting; such datacenters are capable of training far more advanced models (e.g., [3-4]). In this case, that would be diffusion models.

**Questions:**

(1) Given that the initialization is frozen, how robust is PRISM to the random seed? Does this robustness hold across identical hyperparameters?

(2) Could the authors help to contextualize an industrial or academic setting in which PRISM may be deployed in practice?

(3) Are 10 clients really enough to evaluate cross-device "non-IID" performance? For example, the authors note that they split MNIST into 40 disjoint partitions based on class labels. I'm assuming this means that each digit has 4 partitions. Then, the 40 portions are sampled uniformly without replacement and 4 each are assigned to a single client. Am I correct to assume the participation percentage is 100% (which would imply cross-silo)? This does imitate label imbalance, but the number of the clients seem far too low. I would have assumed 2 participating clients out of, say, 200 mobile devices to evaluate for non-IID (e.g., [5]). Also, to my knowledge, a more robust and realistic non-IID partitioning strategy in converting centralized datasets to FL datasets is to use LDA partitioning (widely used in topic modeling).

[1] Sparse Random Networks for Communication-Efficient Federated Learning (Isik et al., ICLR 2023)

[2] Can We Find Strong Lottery Tickets in Generative Models? (Yeo et al., AAAI 2023)

[3] DiLoCo: Distributed Low-Communication Training of Language Models (Douillard et al., Arxiv 2023)

[4] Asynchronous Local-SGD Training for Language Modeling (Liu et al., ICML workshop 2024)

[5] Efficient Adaptive Federated Optimization (Lee et al., ICML workshop 2024)

---

> ### Author Response · Authors · 2024-11-20
> **[Response 1/3] Thank you for constructive feedbacks!**
>
> **R4-1. Additional experiments on different random initialization or seed [R4-Q1]**
>
> We conducted additional experiments across various seed values to provide further performance insights. As can be seen in the table below, PRISM shows robust performance regardless of specific initialization value or random seed with identical hyperparameters.
>
> |        MNIST, IID, DP     |     FID     |     P&R     |      D&C       |
> |:---:|:---:|:---:|:---:|
> |  PRISM in paper (seed30)  |  27.3017  | 0.4377 / 0.5576  | 0.1738 / 0.1982 |
> |  PRISM w/ seed123     |  26.8707  | 0.4342 / 0.5399  | 0.174 / 0.2047 |
> |  PRISM w/ seed1234   |  26.9664  |  0.4417 / 0.5842 | 0.1785 / 0.1997 |
>
> |        MNIST, Non-IID, DP     |     FID     |     P&R     |      D&C       |
> |:---:|:---:|:---:|:---:|
> |  PRISM in paper (seed30)  |  34.2038  | 0.4386 / 0.4236  | 0.1734 / 0.1597 |
> |  PRISM w/ seed123     |  34.0586  | 0.4372 / 0.3306  | 0.1732 / 0.1618 |
> |  PRISM w/ seed1234   |  32.9423  |  0.3921 / 0.4182  | 0.1496 / 0.1587 |
>
> **R4-2. Clarification on the comments: “Figures 2/3 for PRISM and baselines look problematic.” & “CIFAR10 and CelebA are not complex datasets"[R4-W2, R4-W3]**
>
> We understand your concern that demonstrating performance on simpler datasets, such as MNIST, may appear less impactful as an empirical result. However, it is important to note that the field of federated generative models remains far behind the current trend of using more complex and powerful models, such as diffusion models, and large-scale datasets. This gap arises from the inherent difficulties in adapting these advanced models to FL setups.
>
> Existing works have struggled to achieve stable performance in challenging setups, such as Non-IID and privacy-preserving cases, even with MNIST-level datasets. Even for centralized differential privacy generative models [1-2],training generative models is highly challenging and often unstable. Due to these obstacles, prior studies have primarily focused on simpler datasets, such as MNIST, and have been largely restricted to GANs. However, even under these conditions, many of these works failed to achieve desirable results. We summarize the FID scores and experimental setups reported in these works.
>
> |    FID, IID case  |    MD-GAN [3]   |    UA-GAN [4]   |   Multi-FLGAN [5]   |
> |:---:|:---:|:---:|:---:|
> |          MNIST      |         16.81       |         17.34       |         17.1       |
>
> In contrast, PRISM is the first to consistently demonstrate stable performance under these conditions.
>
> The purpose of our experiments was to highlight the limitations of prior approaches and showcase PRISM’s effectiveness in overcoming these challenges. Nevertheless, we acknowledge your doubts and have included additional experimental results below to further support our claims.
>
> Still, we also recognize the importance of assessing performance on high-resolution and large-scale datasets. Considering computational resources and the rebuttal period, we conducted experiments on the CelebA 128x128 and CIFAR100 datasets under Non-IID conditions without considering privacy. The results are summarized in the table below, with qualitative results provided in Appendix.I. Please note that PRISM is the first to achieve this level of performance under the current setting, whereas existing baselines have struggled even on MNIST-level benchmarks.
>
> |    CelebA 128x128  |     FID      |     P&R      |      D&C       |
> |:---:|:---:|:---:|:---:|
> |  PRISM       |  40.2927  |  0.7738 / 0.006  | 1.00207 / 0.348 |
>
> |       CIFAR100        |     FID     |     P&R     |      D&C       |
> |:---:|:---:|:---:|:---:|
> |  PRISM       |  74.1609  | 0.6655 / 0.0719   |  0.602 / 0.3121 |
>
>
> [1] Dockhorn, Tim, et al. "Differentially private diffusion models." arXiv preprint arXiv:2210.09929 (2022).
>
> [2] Jiang, Zepeng, Weiwei Ni, and Yifan Zhang. "PATE-TripleGAN: Privacy-Preserving Image Synthesis with Gaussian Differential Privacy." arXiv preprint arXiv:2404.12730 (2024).
>
> [3] Hardy, Corentin, Erwan Le Merrer, and Bruno Sericola. "Md-gan: Multi-discriminator generative adversarial networks for distributed datasets." 2019 IEEE international parallel and distributed processing symposium (IPDPS). IEEE, 2019.
>
> [4] Zhang, Yikai, et al. "Training federated GANs with theoretical guarantees: A universal aggregation approach." arXiv preprint arXiv:2102.04655 (2021).
>
> [5] Amalan, Akash, et al. "Multi-flgans: multi-distributed adversarial networks for non-IID distribution." arXiv preprint arXiv:2206.12178 (2022).
>
> ...
>
> Please refer the remaining responses in the [Respones 2/3].

---

> > ### Author Response · Authors · 2024-11-20
> > **[Response 2/3] Thank you for constructive feedbacks!**
> >
> > **R4-3: Response to Concerns About Comparisons with Centralized Settings and Practical Scenarios [R4-W3, R4-Q2]**
> >
> > We respectfully disagree with the premise that comparisons with centralized settings are necessary to validate the contributions of our work. While centralized training on mature datasets may indeed be practicable on modern laptops, the focus of our study lies in addressing the unique challenges posed by federated learning (FL) environments, where data is inherently distributed and cannot be shared due to privacy constraints. These scenarios represent practical and increasingly relevant use cases, especially as the demand for privacy-preserving machine learning solutions grows.
> >
> > For example, federated training setups are highly applicable in real-world scenarios such as healthcare, where sensitive medical image data is distributed across multiple institutions, and user data on mobile devices, such as iPhones, where privacy concerns prevent centralized aggregation. Our work addresses these restrictive scenarios by demonstrating stable and effective generative model training under challenging Non-IID and privacy-preserving conditions, where centralized baselines are not directly applicable.
> >
> > Thus, rather than comparing against centralized models, our contribution lies in advancing federated generative modeling techniques to operate effectively in such distributed and restrictive environments. We hope this clarifies the motivation and significance of our work.
> >
> > **R4-4. Additional experiments**
> >
> > **(a) Number of clients>10[ R4-W5]**
> >
> > We believe that the efficiency of PRISM makes it applicable across a wide range of scenarios. In fact, the settings used in our experiments were designed to just establish an intuitive FL environment rather than targeting specific scenarios (e.g., cross-silo or cross-device). Given the restricted resource budget during the rebuttal period, as an initial step to evaluate PRISM’s performance in resource-restricted cross-device settings [1], we conducted additional experiments involving 50 clients, where 10 clients (i.e., 0.2 ratio) participating in each communication round. The results, presented below, show that PRISM maintains robust performance in these cross-device environments, further showcasing its versatility and effectiveness across diverse federated learning scenarios.
> >
> > |        MNIST, Non-IID, DP              |     FID     |     P&R     |      D&C       |
> > |:---:|:---:|:---:|:---:|
> > |  DP-FedAvgGAN |  118.3975  |  0.1095 / 0.3723  |  0.0301 / 0.0289  |
> > |  GS-WGAN    |  98.6563  | 0.8477 / 0.0359  | 0.2621 / 0.0105  |
> > |  PRISM       |  34.7157 | 0.4344 /0.3401  | 0.1692 / 0.1476  |
> >
> > |        MNIST, Non-IID, No-DP              |     FID     |     P&R     |      D&C       |
> > |:---:|:---:|:---:|:---:|
> > |  MD-GAN         |  15.4119  | 0.7305 / 0.359 | 0.5266 / 0.3803  |
> > |  PRISM       |  14.3168 | 0.7533 / 0.4757 | 0.5804 / 0.5293 |
> >
> > **(b) Non-IID [R4-Q3]**
> > In the table below, we report the additional experimental results, Non-IID splitting by Dirichlet distribution with $\alpha=0.005$. Again, we would like to highlight that 1) Other baselines struggle with the 4 shards Non-IID dataset, and 2) PRISM consistently outperforms other baselines under heterogeneity.
> >
> > |        MNIST, Non-IID, DP         |     FID     |     P&R     |      D&C       |
> > |:---:|:---:|:---:|:---:|
> > |  DP-FedAvgGAN |  175.3729  |  0.0408 / 0.1982  | 0.0102 / 0.0048 |
> > |  GS-WGAN   |  128.4401  |  0.0851 / 0.0633  |  0.0196 / 0.0071 |
> > |  PRISM       |  58.7524   | 0.3088 / 0.201  | 0.1078 / 0.0788  |
> >
> > |        MNIST, Non-IID, No-DP         |     FID     |     P&R     |      D&C       |
> > |:---:|:---:|:---:|:---:|
> > |  MD-GAN         |  61.9427  |  0.4292 / 0.1639  | 0.1643 / 0.0747 |
> > |  PRISM       |  31.6191  | 0.5871 / 0.36  |  0.2828 / 0.2328  |
> >
> > [1] Isik, Berivan, et al. "Sparse random networks for communication-efficient federated learning." arXiv preprint arXiv:2209.15328 (2022).
> >
> > ...
> >
> > Please refer the remaining responses in the [Respones 3/3].

---

> ### Author Response · Authors · 2024-11-20
> **[Response 3/3] Thank you for constructive feedbacks!**
>
> **R4-6. Difference between PRISM and prior works [R4-W1]**
>
> We acknowledge that there is some overlap with prior works. However, as R3 also noted, the value of our work lies in effectively combining these methods to address previously unsolved challenges in federated learning for generative models. PRISM demonstrates impressive performance in scenarios where existing baselines fail, successfully tackling many of the prevailing difficulties in this field. In addition, we believe MADA, which is introduced for the first time in our work, constitutes a meaningful contribution by significantly improving performance and providing a novel approach to tackling heterogeneity in federated learning environments. As observed in our experiments, MADA consistently improved performance across datasets, with the improvement being more pronounced under non-IID conditions compared to IID conditions. For example, the FID scores for Non-IID setups showed substantial improvement after applying MADA:
>
> * **MNIST**: Before: 49.6273 → After: 34.2038
> * **FMNIST**: Before: 83.0481 → After: 67.1648
> * **CelebA**: Before: 59.4877 → After: 39.7997
>
> These improvement is even greater than those observed under IID conditions:
>
> * **MNIST**: Before: 48.5636 → After: 27.3017
> * **FMNIST**: Before: 54.722 → After: 46.1652
> * **CelebA**: Before: 57.0573 → After: 48.9983
>
> These results suggest that MADA is effective in addressing the challenges posed by both IID and non-IID settings.
>
> We hope the reviewer recognizes these contributions and the advancements they represent in this domain.
>
> **R4-7. Details in architecture and hyperparameters of PRISM [R4-W4]**
>
> Thank you for your constructive comment. We used a ResNet-based decoder and empirically observed that setting the local epoch to 100 and learning rate to 0.1 effectively optimizes the local stochastic binary masks. All of the detailed instructions have been added to Appendix.C, highlighted in “green”.

---

> > ### Comment · Reviewer_EU7U · 2024-11-21
> > **Thanks for the additional experiments and rebuttals**
> >
> > Thank you for your responses, and your additional experiments. I find your rebuttals quite convincing, and I'm especially encouraged by the extra experiment results you provided. But I would like to engage the authors in some more discussions during the rebuttal period.
> >
> > **[Companions to Centralized Settings]** Clearly, comparing the performance of DP-FL to centralized settings is unfair (as both frameworks have fundamentally different purposes), and FL is known to be upper bounded by a centralized barrier. My point was not that DP-FL needed to be as good as centralized settings in order for there to be a contribution. Rather, my worry is that the frameworks in the paper cannot even generate MNIST digits properly. As I said in my review, "...the authors state that they do not include comparisons with centralized settings as they study settings in which "distributed samples cannot be shared". I think the concern here is that the evaluated datasets are so mature, and generative central training on them easily practicable on even modern Laptops, such that if the evaluated methods cannot even beat the central model, it is difficult to see the contribution being made. Again, my assessment would differ significantly if this was a theoretical paper, but this paper is mainly empirical." Similarly, I am not convinced that statements such as CIFAR10 and CelebA are "complex datasets" which allows evaluation of "state-of-the-art image generation" in line 083 are quite accurate, as I noted in my original review. If the authors could further discuss this point, that would be great.
> >
> > **[On Hospitals and iPhones]** Just quickly as a digression (not important): I find these examples quite interesting, but those examples, along with financial institutions, are standard examples for motivating FL. I'm still trying to contextualize your work in real-world settings. Could the authors clarify how their work can be useful in the context of the examples they provide? Would they be generating hospital or private user data?
> >
> > **[Cross-Device verses Cross-Silo]** From the response of the authors, I understand that the authors are currently in a compute-restricted environment. I believe that this should be taken into consideration. Thank you for the experiments for "Number of Clients > 10"; I would be satisfied if these can be included in the next draft of the paper.
> >
> > **[Learning Rate]** My original enquiry was trying to understand if sufficient hyperparameter tuning had been done for the other baselines to showcase their best performance, in order to be fair to all algorithms being compared. I'd like to ask how this was achieved.

---

> ### Author Response · Authors · 2024-11-21
> **[Response 1/3] Thank you for your prompt feedback!**
>
> **R4-8. Response to Concerns About Dataset and Experimental Setup [R4-W3]**
>
> We sincerely appreciate your feedback, which reflects a clear understanding of the fundamental principles of FL setups. However, we were unsure about certain aspects of your comment and would greatly appreciate any clarification regarding your concerns. You reference our statement that "distributed samples cannot be shared" to explain why centralized comparisons were excluded, but also suggest that our datasets are "mature and simple" enough for centralized training on modern laptops. This seems to conflict with the assumptions of the FL setup, where the full dataset is not accessible at each device, requiring collaboration among the clients to build a reliable model. Furthermore, your statement that "if the evaluated methods cannot even beat the central model, it is difficult to see the contribution being made" adds further ambiguity. Are you concerned that PRISM underperforms in centralized setups where the full dataset is accessible? If so, Appendix G demonstrates that PRISM performs well in such scenarios. Here, if a single laptop is assumed to have access to the entire dataset (e.g., full MNIST dataset), this scenario is considered as a centralized setting, unlike the FL scenario where each client has access to only a limited number of data samples.
>
> First, to clarify your question, "Why do PRISM and the baselines in the paper fail to properly generate MNIST digits in Figures 2 and 3?" the observed challenges arise not from limitations of PRISM itself but from the stringent constraints imposed by the FL+DP (Federated Learning with Differential Privacy) setup we model. While generating MNIST digits is indeed trivial in centralized setups—even on a single laptop—this task becomes highly challenging under FL+DP conditions [1-4]. For example, as shown in Figure 4 of [1], generating MNIST digits while adhering to privacy constraints remains a significant challenge, even in centralized settings.
>
> Second, regarding the evaluated datasets, we would like to emphasize that MNIST, CIFAR-10, and similar benchmarks are commonly used in FL literature to simulate challenging and constrained scenarios with limited data at each client and non-IID data distribution [5-7]. While it is true that these datasets can be fully utilized on a laptop for centralized training, the FL paradigm assumes that data is inherently distributed across multiple devices, and full datasets cannot be shared. This distinction is key: FL setups explicitly model environments where no single device or server has access to the entire dataset, which aligns with real-world applications. Thus, using subsets of datasets such as MNIST and CIFAR-10 is a standard benchmarking practice to evaluate methods under these constraints.
>
> ...
>
> Please refer the following responses [Response 2/3]

---

> > ### Author Response · Authors · 2024-11-21
> > **[Response 2/3] Thank you for your prompt feedback!**
> >
> > Additionally, we model an even more challenging scenario by incorporating privacy-preserving mechanisms, further amplifying the difficulty of the task. Under these stringent conditions, prior works have consistently struggled to achieve meaningful results, particularly on Non-IID distributions [8-11]. In contrast, PRISM demonstrates significantly better performance in these settings, as evidenced by the experimental results presented in the manuscript.
> > Lastly, regarding the phrase "state-of-the-art image generation on complex datasets" in line 083, we acknowledge that it could be perceived as somewhat ambiguous. What we intended to convey is that PRISM achieves state-of-the-art performance within the constrained FL+DP setup. To avoid any misunderstanding, we will revise the manuscript to clarify that this claim pertains specifically to the FL+DP setup and the associated challenges.
> >
> > P.S. We do recognize that the back-and-forth nature of written exchanges can sometimes lead to misunderstandings. While we are confident in the FL benchmarking practices and the relevance of our experiments, we are happy to engage in further discussion with patience and an open mind to better address any remaining concerns or clarify potential misunderstandings.
> >
> > Thank you again for your effort.
> >
> > [1] Jiang, Zepeng, Weiwei Ni, and Yifan Zhang. "PATE-TripleGAN: Privacy-Preserving Image Synthesis with Gaussian Differential Privacy." arXiv preprint arXiv:2404.12730 (2024).
> > [2] Zhang, Yikai, et al. "Training federated GANs with theoretical guarantees: A universal aggregation approach." arXiv preprint arXiv:2102.04655 (2021).
> > [3] Li, Wei, et al. "Ifl-gan: Improved federated learning generative adversarial network with maximum mean discrepancy model aggregation." IEEE Transactions on Neural Networks and Learning Systems 34.12 (2022): 10502-10515.
> > [4] Xin, Bangzhou, et al. "Private fl-gan: Differential privacy synthetic data generation based on federated learning." ICASSP 2020-2020 IEEE International Conference on Acoustics, Speech and Signal Processing (ICASSP). IEEE, 2020.
> > [5] Tang, Yitong. "Adapted Weighted Aggregation in Federated Learning." Proceedings of the AAAI Conference on Artificial Intelligence. Vol. 38. No. 21. 2024.
> > [6] Ji, Xinyuan, et al. "FedFixer: Mitigating Heterogeneous Label Noise in Federated Learning." Proceedings of the AAAI Conference on Artificial Intelligence. Vol. 38. No. 11. 2024.
> > [7] Rahimi, Mohammad Mahdi, et al. "EvoFed: leveraging evolutionary strategies for communication-efficient federated learning." Advances in Neural Information Processing Systems 36 (2024).
> > [8] Chen, Dingfan, Tribhuvanesh Orekondy, and Mario Fritz. "Gs-wgan: A gradient-sanitized approach for learning differentially private generators." Advances in Neural Information Processing Systems 33 (2020): 12673-12684.
> > [9] Augenstein, Sean, et al. "Generative models for effective ML on private, decentralized datasets." arXiv preprint arXiv:1911.06679 (2019).
> > [10] Hardy, Corentin, Erwan Le Merrer, and Bruno Sericola. "Md-gan: Multi-discriminator generative adversarial networks for distributed datasets." 2019 IEEE international parallel and distributed processing symposium (IPDPS). IEEE, 2019.
> > [11] Amalan, Akash, et al. "Multi-flgans: multi-distributed adversarial networks for non-IID distribution." arXiv preprint arXiv:2206.12178 (2022).
> >
> > ...
> >
> > Please refer the following responses [Response 3/3]

---

> > > ### Author Response · Authors · 2024-11-21
> > > **[Response 3/3] Thank you for your prompt feedback!**
> > >
> > > **R4-9. On Hospitals and iPhones**
> > >
> > > Thank you for your follow-up question. We agree that examples such as hospitals, financial institutions, and mobile devices are standard scenarios for motivating FL. In the context of our work, the primary focus is not on generating hospital or private user data directly, but on developing methodologies that enable stable and effective generative modeling under the constraints typically faced in these settings. The trained generative model can be used for various use-cases, as we describe below.
> > >
> > > For instance, in healthcare, our approach could be used to collaboratively train generative models across multiple institutions without sharing sensitive medical data. These generative models could then support downstream tasks such as synthetic data generation for privacy-preserving research, imputation of missing patient data, data augmentation to improve model robustness in low-data scenarios, or providing priors for solving inverse problems (e.g., denoising, reconstruction, super-resolution, etc.). Similarly, in mobile devices, the ability to train generative models across distributed devices while preserving user privacy could enhance applications like personalized content generation or user-specific data augmentation.
> > >
> > > While federated generative models in general may address these challenges to some extent, PRISM offers several key advantages. Beyond achieving superior performance under challenging Non-IID and privacy-preserving conditions, PRISM also improves communication efficiency and security by transmitting only sparse masks instead of full weights or gradients. As discussed in Appendix B of the main paper, this approach significantly mitigates potential privacy risks associated with transmitting generative model weights, which could be exploited to reproduce private data.
> > >
> > > For example, even if a malicious actor intercepts the sparse masks during communication, these masks alone are insufficient to reconstruct the original data unless the attacker also knows the exact model architecture and the randomly initialized weights used in each local device. This is a notable contrast to methods that transmit full weights or gradients, which pose higher risks of privacy leakage once the model details are exposed.
> > >
> > > Furthermore, PRISM enhances privacy protection even against the server, as the server only aggregates and returns masks without needing access to sensitive weight information. This differs fundamentally from approaches that require sending complete weights or gradients to the server, which inherently carry more privacy-sensitive information.
> > >
> > > We hope this clarifies how our work aligns with the practical scenarios provided and how PRISM contributes uniquely to addressing these challenges.
> > >
> > >
> > > **R4-10. Cross-Device verses Cross-Silo**
> > > Your comment on cross-device scenarios has been invaluable in strengthening our paper. We include these experimental results and discussion in the Appendix K (highlighted in “green”) to help readers better understand the applicability of our method across various scenarios.
> > >
> > >
> > > **R4-11. Learning Rate**
> > >
> > > We did our best to ensure a fair comparison. To this end, we conducted experiments using the hyperparameter values reported in prior works or by using the official code provided by the authors. These hyperparameter settings are, in fact, those optimized by the original authors for the same datasets we used, representing their best possible configurations. For GS-WGAN, we even followed the process of warming up the discriminator through pretraining.
> > > Nevertheless, to further address your concern regarding the fairness of our experimental process, we plan to conduct additional experiments on various configurations, such as learning rate and local epochs. Detailed results will be shared as they become available to keep the reviewers informed.

---

> > ### Comment · Reviewer_EU7U · 2024-11-21
> >
> > Thanks a lot for your reply--it is very much appreciated. I also appreciate the revisions you are making.
> >
> > **[MNIST generation]** Just quickly, regarding the first point: I mentioned previously that "my worry is that the frameworks in the paper cannot even generate MNIST digits properly." To clarify the "ambiguity" that you mentioned, this was my main concern, as to my understanding we would expect papers published in ICLR to be able to generate MNIST digits properly. In your reply, you clarified quite well that in the setting that is analyzed in PRISM, this was not a trivial task due to the additional complexities imposed by DP, FL, masking, etc. Given this context, and the additional experiments you have provided, I am willing to be convinced that your algorithm presents a reasonable contribution. Point well taken; thanks for providing the additional works.
> >
> > **[Learning Rate]** Thanks for clarifying this. I'll take your word that you'll conduct and add additional experiments for verifying the fairness of your experimental process.
> >
> > Thanks for resolving my enquiries and comments about your work. I'll think about this some more and then revise my score accordingly.

---

> > > ### Comment · Reviewer_EU7U · 2024-11-21
> > >
> > > I have updated my score.

---

> ### Author Response · Authors · 2024-11-24
> **[Response 1/1] Appreciate the reviewer for acknowledging our efforts and raising the score.**
>
> We appreciate the reviewer for acknowledging our efforts and raising the score.
>
> Below, we present additional experiments conducted with different learning rates in response to the reviewer’s comment.
>
> **R4-12. Explore various learning rates for baselines**
>
> To validate the fairness of our experiments, we explored various configurations of learning rates for the baselines. Since they utilize GANs, there are separate learning rates for the generator and discriminator, and we examined these combinations. In the tables below, the top row represents the discriminator’s lr / generator’s lr, while the subsequent row reports the FID scores. Due to the notorious instability of GAN training caused by the coupled learning dynamics between the generator and discriminator [1-3], we observed divergence in some settings. For GS-WGAN, the best performance was achieved with 1e-4 / 1e-5, but it still failed to generate MNIST images properly. We include these results on Appendix L in our main paper.
>
> |      MNIST, Non-IID, DP    |   1e-4, 1e-4 (default)   |     1e-3, 1e-4   |     1e-5, 1e-4   |     1e-4, 1e-3   |     1e-4, 1e-5   |
> |:---:|:---:|:---:|:---:|:---:|:---:|
> |  GS-WGAN   |  diverge  | 108.0657 | diverge | diverge | 96.1892
>
> |      MNIST, Non-IID, DP    |   1e-3, 5e-4 (default)  |     1e-3, 5e-3   |     1e-2, 5e-4   |     1e-4, 5e-4   |     1e-3, 1e-5   |
> |:---:|:---:|:---:|:---:|:---:|:---:|
> |  DP-FedAvgGAN  |  153.9325 | 206.759 | diverge | 177.3752 | 232.2336 |
>
> [1] Miyato, Takeru, et al. "Spectral normalization for generative adversarial networks." arXiv preprint arXiv:1802.05957 (2018).
> [2] Mescheder, Lars, Andreas Geiger, and Sebastian Nowozin. "Which training methods for GANs do actually converge?." International conference on machine learning. PMLR, 2018.
> [3] Mao, Xudong, et al. "Least squares generative adversarial networks." Proceedings of the IEEE international conference on computer vision. 2017.

---

### Official Review · Reviewer_NWco · 2024-10-31

**Soundness:** 3
**Presentation:** 4
**Contribution:** 2
**Rating:** 6
**Confidence:** 3

**Summary:**

The authors propose a federated masking method for generative models that reduces communication and memory costs as well as improves performance in non-iid settings. A binary mask over the entire model is selected via the Strong Lottery Ticket (SLT) hypothesis, by which a maximum mean discrepancy (MMD) loss is used to provide gradient feedback to update weight-importance scores. Devices use these scores to generate (via Bernoulli distribution) a binary mask that is passed to the server for aggregation (thereby reducing communication costs). Finally, the server performs mask-aware dynamic moving average aggregation (MADA) whereby the server determines the drift of the global model and uses the value to determine how much of the new mask update should be incorporated. Experimental results show large increases in performance metrics while reducing communication and memory costs.

**Strengths:**

The strongest aspect are the empirical results of the proposed method. PRISM outperforms all of the baselines by a good margin while also being more efficient (in communication and memory). This is impressive.

The proposed MADA method is an interesting idea to alleviate model drift and improves performance empirically. As detailed in the questions, it seems to improve performance overall but not alleviate the effects of non-iid settings.

The paper writing and presentation is great. I could follow easily, the diagram is very helpful and the empirical results (both tables and figures) are presented cleanly. Great job!

**Weaknesses:**

I do not believe that the PRISM method itself is especially novel or specifically tailored for generative models as the authors market it as. The major performance driver behind PRISM seems to be the idea of training a sparse binary mask. This is shown to be quite effective in FedMask (Li et a. 2021), and it makes sense why it would be effective on GANs as well. However, this seems to be a simple application of FedMask coupled with the MMD loss (used in GAN training) and the WADA update to GANs. In fact, outside of using MMD loss, I do not see how PRISM is tailored towards generative models. The entire process could be applied to any other ML models if the MMD loss, which has already been proposed before, is substituted out.

Communication costs only seem to be saved during the uplink process, when only the binary mask is sent to the server. However, the downlink communication cost remains the same since the learnable weight-importance scores must be sent down to all devices. This is not mentioned by the authors and thus does not tackle all the communication issues.

The non-iid empirical setting seems slightly odd. Namely the authors partition "datasets into 40 segments based on class labels and randomly assign four segments to each client". Usually a Dirichlet split is the best way to simulate a non-iid split of labels amongst classes. As a result, it is odd that the performance of all algorithms stays around the same or *improves* for certain metrics. Precision and Recall improves for some of the other baselines which do not account for non-iidness. Generally non-iid settings should degrade performance in FL settings and that is not the case here for all baselines. This makes me feel that the partitioning was not non-iid enough.

**Questions:**

Is MADA derived from other literature, or is this proposed here for the first time?

Could the authors further detail the communication savings of PRISM? It seems that there is no downlink communication savings?

I was a bit confused about the model memory savings (Line 242). Could the authors clarify this? What I took away is that since model weights are frozen, and since they are initialized using the Kaiming Normal distribution, only the -/+ bit needs to be saved? Is this where the memory savings stem from (as shown in Tables 1/2 and Figure 4)?

The authors mention in Line 276:"As the global rounds progress, λ gradually decreases, promoting stable convergence." Is this due to model convergence arising from learning rate decay? What I mean is that, especially in non-iid settings, the global model generally converges once the learning rate decays and thus this should in turn decrease λ. Is that the correct characterization?

Could the authors provide new non-iid experiments that showcase a more realistic setting?

Overall, I feel that the application of sparse binary mask training to GANs is powerful and effective as shown in this paper. While each component is not especially novel, the combination suited for GANs is a novelty (albeit not a major one). However, the empirical performance is impressive and I believe in conjunction leads to a nice paper. The only disclaimer is that I am not especially well-versed within GAN or FL sparse binary mask literature.

---

> ### Author Response · Authors · 2024-11-20
> **[Response 1/2] Thank you for constructive feedbacks!**
>
> **R3-1. Further explanation of MADA**
>
> **(1) Is MADA derived from other literature, or is this proposed here for the first time? [R3-Q1]**
>
> To the best of our knowledge, MADA, which determines the moving average ratio on the server side based on the similarity of the global model, is proposed here for the first time.
>
> **(2) Is MADA convergence due to learning rate decay? [R3-Q4]**
>
> First, we would like to clarify that we did not use learning rate decay throughout the FL process. Thus, the local model continues to update consistently using a constant learning rate. During the FL process, the decrease in $\lambda$ is due to the convergence of the binary mask, which reduces the similarity between the two global models, rather than being a result of learning rate decay.
>
> **R3-2. Communication cost savings of PRISM [R3-Q2, R3-W2]**
>
> PRISM transmits a float-type score during downlink, which, as you noted, does not support efficient communication, unlike the binary mask transmission in the uplink. In a typical FL setup, the server possesses powerful computational capabilities, whereas the clients do not. As a result, several prior studies primarily focus on addressing the limited bandwidth of clients [1-3]. Accordingly, we have concentrated on the communication cost in the downlink. We will clarify this in our manuscript (Appendix.J, highlighted in “lime”).
>
> **R3-3. Memory savings of frozen model [R3-Q3]**
>
> Our memory-saving approach is akin to ternary quantization [4], a technique that quantizes neural network weights to {1, 0, -1} through thresholding and projection, making the sign the critical factor in the process. Similar but distinct, PRISM's frozen weights are already initialized to signed constant using Kaiming normal distribution standard deviation, $\sigma_l$. Consequently, after FL concludes, each client can achieve additional memory savings by storing not only the lightweight binary mask but also the frozen weights with the scaling factor {$\sigma_1$, …, $\sigma_l$} and a 1-bit value representing the sign. We hope our clarification resolves any confusion, but let us know if further explanation is needed.
>
>
> **R3-4. Clarification on Non-IID Empirical Settings and Additional experiments with Dirichlet Non-IID split [R3-W3, R3-Q5]**
>
> As suggested, we conducted additional experiments under Non-IID splitting by Dirichlet distribution ($\alpha=0.005$). We report the detailed results in the table below. PRISM continues to outperform other baselines in both Non-IID with/without DP setups, demonstrating its robustness.
>
> |        MNIST, Non-IID, DP         |     FID     |     P&R     |      D&C       |
> |:---:|:---:|:---:|:---:|
> |  DP-FedAvgGAN |  175.3729  |  0.0408 / 0.1982  | 0.0102 / 0.0048 |
> |  GS-WGAN   |  128.4401  |  0.0851 / 0.0633  |  0.0196 / 0.0071 |
> |  PRISM       |  58.7524   | 0.3088 / 0.201  | 0.1078 / 0.0788  |
>
> |        MNIST, Non-IID, No-DP         |     FID     |     P&R     |      D&C       |
> |:---:|:---:|:---:|:---:|
> |  MD-GAN         |  106.3468  |  0.4292 / 0.105  | 0.1643 / 0.0332 |
> |  PRISM       |  31.6191  | 0.5871 / 0.36  |  0.2828 / 0.2328  |
>
> We would also like to address the concern (potential misunderstanding) regarding the Precision and Recall (P&R) results in our original experiments. While FID scores, which are widely regarded as a reliable metric for generative models, exhibit clear trends under non-IID settings, interpreting P&R and D&C metrics requires caution due to their inherent characteristics. For instance, fidelity metrics like Precision can yield inflated values under mode collapse, where many generated samples concentrate around a single real sample. Similarly, outliers in the generated data can disproportionately impact P&R scores, leading to variability and potentially misleading results, as highlighted in prior studies [5]. These characteristics can explain why P&R and D&C results appear similar to or even exceed those of IID settings, despite being evaluated under a Non-IID scenario.
>
> ...
>
> remaining responses will be posted soon.
>
> [1] Kim, Do-Yeon, et al. "Achieving Lossless Gradient Sparsification via Mapping to Alternative Space in Federated Learning." Forty-first International Conference on Machine Learning.
>
> [2] Hu, Rui, Yuanxiong Guo, and Yanmin Gong. "Federated learning with sparsified model perturbation: Improving accuracy under client-level differential privacy." IEEE Transactions on Mobile Computing (2023).
>
> [3] Yi, Liping, Wang Gang, and Liu Xiaoguang. "QSFL: A two-level uplink communication optimization framework for federated learning." International Conference on Machine Learning. PMLR, 2022.
>
> [4] Liu, Dan, and Xue Liu. "Ternary Quantization: A Survey." arXiv preprint arXiv:2303.01505 (2023).
>
> [5] Naeem, Muhammad Ferjad, et al. "Reliable fidelity and diversity metrics for generative models." International Conference on Machine Learning. PMLR, 2020.
>
> ...
>
> Please refer the remaining responses in the [Respones 2/2].

---

> > ### Author Response · Authors · 2024-11-20
> > **[Response 2/2] Thank you for constructive feedbacks!**
> >
> > **R3-5. Addressing PRISM's contributions [R3-W1, R3-Q6]**
> >
> > First, thanks for your positive feedback on our work. While we acknowledge that some components overlap with prior works, as you noted, the value of our work lies in effectively combining these methods and successfully addressing previously unsolved challenges in federated learning for generative models. In addition, we would like to highlight the proposed MADA framework, which demonstrates significant performance improvements. We believe this represents a noteworthy contribution to mitigating heterogeneity in federated learning environments.
> >
> >
> > Regarding your comment that MADA “seems to improve performance overall but not alleviate the effects of non-iid settings,” we are curious about what led you to this interpretation. Based on our results, MADA consistently improved performance across datasets, with the improvement being more pronounced under non-IID conditions compared to IID conditions. For example, the FID scores for Non-IID setups showed substantial improvement after applying MADA:
> >
> > * **MNIST**: Before: 49.6273 → After: 34.2038
> > * **FMNIST**: Before: 83.0481 → After: 67.1648
> > * **CelebA**: Before: 59.4877 → After: 39.7997
> >
> > These improvement is even greater than those observed under IID conditions:
> >
> > * **MNIST**: Before: 48.5636 → After: 27.3017
> > * **FMNIST**: Before: 54.722 → After: 46.1652
> > * **CelebA**: Before: 57.0573 → After: 48.9983
> >
> > These results suggest that MADA is effective in addressing the challenges posed by non-IID settings. We would appreciate any additional insights you might have on this point, as understanding your perspective could help us further refine our work.

---

> > > ### Comment · Reviewer_NWco · 2024-11-21
> > > **Thanks for the Detailed Responses!**
> > >
> > > Dear Authors,
> > >
> > > Thank you for the detailed responses.
> > >
> > > I reaffirm that I believe that the paper effectively combines multiple methods in order to achieve impressive empirical performance.
> > >
> > > > Regarding your comment that MADA “seems to improve performance overall but not alleviate the effects of non-iid settings,” we are curious about what led you to this interpretation.
> > >
> > > What I was mentioning within my review was that you mention "As the global rounds progress, λ gradually decreases, promoting stable convergence" in Line 276. However, this does not seem to be a given in non-IID settings. There likely will be model drift that occurs that will cause the difference between consecutive global masks to not necessarily lead to a gradual reduction in $\lambda$. As a result, I was inquiring how exactly $\lambda$ decreases in these heterogeneous environments. My possible explanation was a reduction in learning rate. Does that make sense?
> > >
> > > Overall, I think MADA provides an interesting technique to fight model drift, however I am unsure what theoretical underpinnings it has. Could the authors expand on that?
> > >
> > > > In a typical FL setup, the server possesses powerful computational capabilities, whereas the clients do not.
> > >
> > > I understand what the authors are arguing, and I don't believe this is a huge drawback. However, it does require smaller devices to receive a very large model (which could be memory-infeasible). Furthermore, if there are hundreds of thousands of devices, the amount of communication could be quite difficult for the server. Again, I understand that it's better to have compression in one way than none. However, the downlink process is still important.
> > >
> > > > Memory savings of frozen model [R3-Q3]
> > >
> > > I may have to read through this paper. I am still a little confused. Can these models really still perform super well with only 1-bit?
> > >
> > > > As suggested, we conducted additional experiments under Non-IID splitting by Dirichlet distribution ($\alpha=0.005$).
> > >
> > > I appreciate the added experimental results. Not to be a pain, but the issues I mentioned in the rebuttal namely pertained to the CelebA dataset, as the Non-IID performance seemed to improve on that dataset compared to the IID setting. This was confusing and made the results a little murkier.
> > >
> > > Best,
> > >
> > > Reviewer NWco

---

> > > > ### Author Response · Authors · 2024-11-24
> > > > **[Response 1/2] Response to Reviewer NWco**
> > > >
> > > > Thank you for carefully reviewing our paper and specifying your question again. We are trying to best address your concerns.
> > > >
> > > > **R3-6. Detailed explanation about MADA**
> > > >
> > > > **[Regarding learning rate]**
> > > >
> > > > Given the local model $w_k^t = w_k^{t-1}-\eta \cdot \Delta_k^t$ where $\eta$ is constant learning rate and $\Delta_k^t$ is local model update, the global model in FedAvg can be obtained as $w^t_{FedAvg} = w_k^{t-1}-\eta \cdot \sum_{k}\Delta_k^t$. In the case of MADA, the global model can be represented as $w^t_{MADA} = (1-\lambda) w_k^{t-1}+\lambda\cdot w_k^t$, where $w_k^t = w_k^{t-1}-\eta \cdot \Delta_k^t$.
> > > >
> > > >  Substituting this into the equation:
> > > >
> > > > $w^t_{MADA}=(1-\lambda) w_k^{t-1}+\lambda\cdot (w_k^{t-1}-\eta \cdot \sum_{k}\Delta_k^t) = w_k^{t-1}-\eta \cdot\lambda \sum_{k}\Delta_k^t$, indicating that $\lambda$ effectively modulates the learning rate for global updates. Based on this interpretation, MADA can be considered as an adaptive lr scheduler on the server side by estimating local divergence using similarity. Unlike traditional schedulers with fixed decay, MADA adapts dynamically to FL dynamics, either increasing or decreasing as needed. This flexibility eliminates the need for client-side learning rate scheduling to regularize local objectives.
> > > >
> > > > **[Decreasing $\lambda$ in Non-IID]**
> > > >
> > > > The intuition behind MADA is that model parameters can change drastically because the algorithm begins to minimize the loss function starting from the initialized model. The norm of gradients could be typically large as the model is far from the optimum. Hence, the models at consecutive epochs could differ significantly, leading to a large $\lambda$ [1-2].
> > > >
> > > > The key question, as you pointed out, is whether this trend holds in heterogeneous scenarios. Your interpretation is certainly valid, a potential corner case exists that maintains $\lambda$ large in heterogeneity scenarios. However, we believe that our interpretation to understanding $\lambda$ is equally plausible and, based on the experimental results (see Figure 6 in the main paper), perhaps more natural.
> > > >
> > > > We will elaborate on this question where client-drift occurs. Even if client-drift is biased toward a specific dominant client, the update difference for the dominant client will naturally decrease from the perspective of global support, while the difference with the local solutions of other clients will increase. This behavior is reflected in the aggregated model, and the difference between consecutive global models can partially capture this behavior (e.g., indicated by large $\lambda$). Thus, as the training progresses and the model moves closer to the global optimum of the local objectives, the differences between consecutive global models become smaller, leading to a reduction in $\lambda$. Otherwise, when client-drift does not occur and the model learns fairly across all clients (e.g., IID scenario), it will naturally converge without significant increases in differences.
> > > >
> > > > In summary, MADA updates the global model in a way that avoids local divergence, gradually reducing global model differences even without learning rate decay.  As shown in Figure 6 of our main paper, $\lambda$ decreases sharply in the early stages of training due to significant local divergence but gradually converges over time. This result suggests that our understanding aligns with the behavior of $\lambda$.
> > > >
> > > > If there are any points of our response that you find unclear or if you require more detailed guidance, please let us know so that we can support your understanding.
> > > >
> > > >
> > > > [1] Cao, Yanzhao, Somak Das, and Hans‐Werner van Wyk. "Adaptive stochastic gradient descent for optimal control of parabolic equations with random parameters." Numerical Methods for Partial Differential Equations 38.6 (2022): 2104-2122.
> > > >
> > > > [2] Chatterjee, Sourav. "Convergence of gradient descent for deep neural networks." arXiv preprint arXiv:2203.16462 (2022).
> > > >
> > > > ...
> > > >
> > > > Please refer the remaining responses in the [Respones 2/2].

---

> > > > > ### Author Response · Authors · 2024-11-24
> > > > > **[Response 2/2] Response to Reviewer NWco**
> > > > >
> > > > > **R3-7. Communication cost during downlink process**
> > > > >
> > > > > We agree that downlink cost is also an important consideration, and various studies have focused on compressing it. Thanks to your feedback, the paper has become much clearer. We truly appreciate your insightful comments.
> > > > >
> > > > > **R3-8. Detailed explanation about memory savings of frozen model**
> > > > >
> > > > > Thank you for your question. We will do our best to address your confusion.
> > > > > First, we would like to clarify the concept of ternary quantization [1], to help explain the intuition behind our memory-saving approach. Ternary quantization decomposes $w$ into a scale factor $\alpha$ and signed factor $\hat{w}$ to obtain ternary weights $\lbrace-\alpha, 0, \alpha\rbrace$.
> > > > >
> > > > > Building on this concept, PRISM can further compress the final model. After the FL procedure concludes, each client has a sparse network represented as $W=W_{init} \odot M$. While the binary mask $M$ is a 1-bit array, storing $W_{init}$ can be burdensome on small devices, depending on the model size. PRISM addresses this issue using signed constant initialization, inspired by ternary quantization. Specifically, each layer is initialized as $w_l=\lbrace+\sigma_l, -\sigma_l, …, +\sigma_l\rbrace\in \mathbb{R}^d$, where $\sigma_l \in \mathbb{R}^1$ is a standard deviation from Kaiming Normal distribution. Note that $w_l$ can be decomposed into a signed array $sign_l=\lbrace+1, -1, …, +1\rbrace \in \mathbb{R}^d$ and scale factor $\sigma_l \in \mathbb{R}^1$. Since 1-bit quantized weights require negligible storage, only the scale factors contribute to the memory usage, this results in the lightweight final model size (7.25MB) reported in Tables 1-3 of the main text.
> > > > >
> > > > > **R3-9. Non-IID settings regarding CelebA dataset.**
> > > > >
> > > > > We would like to clarify that CelebA is a multi-label dataset, where each image is associated with multiple attributes (e.g., age, gender, etc.), making it challenging to create an extreme Non-IID setup. As explained in Section 5.2 of the main paper, CelebA dataset was divided into positive and negative partitions for a pivotal attribute (gender, in our case), with the number of partitions corresponding to the number of clients. Each client was assigned either the positive or negative subset, ensuring non-overlapping data for the selected attribute.
> > > > >
> > > > > To further explore this, we conducted additional experiments where each client possessed the pivotal attribute in different proportions using dirichlet distribution. For example, client 1 has 60% male and 40% female, while client 2 has 20% male and 80% female. The results are reported in the table below. To the best of our knowledge, methods for splitting multi-label datasets into extreme Non-IID scenarios have not been explored. If you could share additional insights on this matter, we would be glad to conduct further experiments based on your suggestions.
> > > > >
> > > > > |      CelebA, Non-IID, DP    |     FID     |     P&R     |      D&C       |
> > > > > |:---:|:---:|:---:|:---:|
> > > > > |  PRISM       |  34.2038  | 0.4386 / 0.4236  |  0.1734 / 0.1597 |
> > > > >
> > > > > [1] Zhu, Chenzhuo, et al. "Trained ternary quantization." arXiv preprint arXiv:1612.01064 (2016).

---

### Official Review · Reviewer_1FdJ · 2024-11-03

**Soundness:** 3
**Presentation:** 3
**Contribution:** 3
**Rating:** 6
**Confidence:** 3

**Summary:**

The core idea of this manuscript is to search for the optimal random binary mask of a random network to identify sparse sub-networks with high-performance generative capabilities. By communicating the binary mask randomly, PRISM minimizes communication overhead. Combining maximum mean discrepancy (MMD) loss and a mask-aware dynamic moving average aggregation method (MADA) on the server side, PRISM achieves stable and robust generative capabilities by mitigating local divergence in federated learning scenarios. Moreover, due to its sparse nature, models generated by PRISM are lightweight and well-suited for environments like edge devices without requiring additional pruning or quantization.

**Strengths:**

1. The PRISM framework demonstrates a degree of innovation; although techniques such as the strong lottery ticket hypothesis and MMD are existing methods, the authors have effectively incorporated them into the federated learning setting.

2. In terms of privacy, PRISM alleviates privacy leakage to some extent by introducing an (ϵ, δ)-differential privacy mechanism.

**Weaknesses:**

1. Although PRISM performs well on small-scale datasets, the manuscript does not adequately discuss its scalability and performance on large-scale datasets and larger models.

2. The search process for random binary masks may increase computational complexity, especially in large networks. It is recommended that the authors analyze the specific computational costs of this process.

**Questions:**

None

---

> ### Author Response · Authors · 2024-11-20
> **[Response 1/1] Thank you for constructive feedbacks!**
>
> **R2-1. Scalability and performance on large-scale datasets and larger model [R2-W1]**
>
> We would like to emphasize that recent FL studies often lack discussions on high-resolution and large-scale datasets. As highlighted in the related work section of our main script [3-6] and in recent studies on privacy-preserving generative models [1-2], this reflects the inherent difficulty of training generative models in unstable FL setups. For this reason, prior works have predominantly focused on such experimental environments, with most achieving only MNIST-level generation.
> Nevertheless, we recognize the importance of addressing whether robust performance can be achieved on high-resolution and large-scale datasets. Considering computational resources and the rebuttal period, we conducted experiments on the CelebA 128x128 and CIFAR100 datasets under Non-IID local data distributions without privacy considerations. Quantitative results are reported in the table below, and qualitative results are provided in Appendix I. It is noteworthy that existing baselines have failed not only in these settings but also on MNIST-level benchmarks. We are the first to achieve this level of performance under the current conditions.
> To further address the scalability of PRISM, we will update the main text with relevant discussions and respond accordingly once the content has been revised.
>
> |    CelebA 128x128  |     FID      |     P&R      |      D&C       |
> |:---------------------------:|:--------------:|:---------------:|:--------------------:|
> |  PRISM       |  40.2927  |  0.7738 / 0.006  | 1.00207 / 0.348 |
>
> |       CIFAR100        |     FID     |     P&R     |      D&C       |
> |:---------------------------:|:--------------:|:---------------:|:---------------------:|
> |  PRISM       |  74.1609  | 0.6655 / 0.0719   |  0.602 / 0.3121 |
>
>
>
> **R2-2. Computational cost of SLT process [R2-W2]**
>
> As the number of model parameters increases, the required score parameters also increase linearly. However, in SLT, weight parameters are not learned through gradient descent. The only additional components are the sigmoid and Bernoulli processes introduced during the binary masking procedure. While this does incur some additional computational cost, it is negligible compared to the cost of gradient descent operations on GPUs.
>
> To further substantiate this claim, we are conducting experiments to compare the FLOPS (floating-point operations per second) with and without the binary masking process. Detailed results will be provided as soon as the experiments are completed.
>
> [1] Dockhorn, Tim, et al. "Differentially private diffusion models." arXiv preprint arXiv:2210.09929 (2022).
>
> [2] Jiang, Zepeng, Weiwei Ni, and Yifan Zhang. "PATE-TripleGAN: Privacy-Preserving Image Synthesis with Gaussian Differential Privacy." arXiv preprint arXiv:2404.12730 (2024).
>
> [3] Hardy, Corentin, Erwan Le Merrer, and Bruno Sericola. "Md-gan: Multi-discriminator generative adversarial networks for distributed datasets." 2019 IEEE international parallel and distributed processing symposium (IPDPS). IEEE, 2019.
>
> [4] Zhang, Yikai, et al. "Training federated GANs with theoretical guarantees: A universal aggregation approach." arXiv preprint arXiv:2102.04655 (2021).
>
> [5] Amalan, Akash, et al. "Multi-flgans: multi-distributed adversarial networks for non-IID distribution." arXiv preprint arXiv:2206.12178 (2022).
>
> [6] Li, Wei, et al. "Ifl-gan: Improved federated learning generative adversarial network with maximum mean discrepancy model aggregation." IEEE Transactions on Neural Networks and Learning Systems 34.12 (2022): 10502-10515.

---

> > ### Author Response · Authors · 2024-11-24
> > **Additional Response to Reviewer 1FdJ**
> >
> > **Computation cost analysis with FLOPS**
> >
> > Regarding R2-2, we provide the results of computation cost comparison in the table below. GS-WGAN shows increased FLOPS due to spectral normalization, introduced to stabilize GAN training. In contrast, PRISM, despite performing the SLT process, demonstrates sufficiently low FLOPS.
> >
> > |   DP-FedAvgGAN     |    GS-WGAN     |     PRISM      |
> > |:---:|:---:|:---:|
> > | 0.002 G | 1.94 G |  0.34 G |

---

> > > ### Author Response · Authors · 2024-11-25
> > > **We submit the revised version of manuscript**
> > >
> > > We have included the rebuttal discussions in the appendix, including R2-2, in the appendix and uploaded the revised version. If you have any additional concerns regarding these topics or suggestions for further experiments, please let us know.

---

> > > > ### Author Response · Authors · 2024-12-01
> > > > **Official Comment to Reviewer 1FdJ. We are looking forward to your comment**
> > > >
> > > > Dear Reviewer 1FdJ,
> > > >
> > > > With the author-reviewer discussion period nearing its conclusion, we kindly request your feedback on whether all your concerns have been fully addressed. Should you have any additional questions or require further clarification, please do not hesitate to let us know. We look forward to hearing from you.

---

### Official Review · Reviewer_Aeqb · 2024-11-03

**Soundness:** 3
**Presentation:** 3
**Contribution:** 2
**Rating:** 5
**Confidence:** 3

**Summary:**

This paper introduces PRISM, a federated learning (FL) framework designed specifically for generative models to learn under heterogeneous data and lower the communication costs. The main idea leverages the strong lottery ticket (SLT) hypothesis by identifying an optimal sparse subnetwork with high generative performance through stochastic binary masking. The communication overhead is reduced by exchanging binary masks rather than full model weights. PRISM also includes a mask-aware dynamic moving average aggregation (MADA) to mitigate client drift, and maximum mean discrepancy (MMD) loss for stable training in generative tasks. Experiments demonstrate that PRISM outperforms existing federated generative models, such as GAN trained with DP-FedAvg on different image datasets.

**Strengths:**

- The algorithm is a novel for learning the binary mask in SLT hypothesis under FL setting. The paper is well-written and the included MMD loss and MADA are well-justified approaches to handle data heterogeneity.
- The experiments considered different settings (e.g. IID v.s. non-IID and privacy v.s. no privacy) and the proposed PRISM showed considerable gain compared to the baseline approaches.

**Weaknesses:**

There seems to be limited advances in GAN in the recent generative models literature so I am not sure if the experiment setting of GAN with simple image datasets (e.g. MNIST or CIFAR10) would still be a significant result for the community.

**Questions:**

- What is the baseline performance of purely server-side SLT? E.g. without any FL setup.
- Would the approach work for diffusion-based generative models?

---

> ### Author Response · Authors · 2024-11-20
> **[Response 1/2] Thank you for constructive feedbacks!**
>
> **R1-1. Baseline performance of purely server-side SLT [R1-Q1]**
>
> We understand that your inquiry is about the performance of PRISM under a centralized setting. In Table 6 of Appendix G in the main paper, we compared PRISM (Non-IID and privacy-preserving) with PRISM-vanilla (centralized setting). For your convenience, we present the same results in the table below.
> As shown in prior studies [1-3], centralized SGD typically outperforms federated settings, and our findings align with this trend.
> |       MNIST    |     FID     |     P&R     |      D&C       |
> |:---:|:---:|:---:|:---:|
> |  PRISM       |  34.2038  | 0.4386 / 0.4236  |  0.1734 / 0.1597 |
> |  PRISM (vanilla)   |  5.8238  | 0.6913  / 0.851  |  0.4689 / 0.679 |
>
> |       FMNIST    |     FID     |     P&R     |      D&C       |
> |:---:|:---:|:---:|:---:|
> |  PRISM       |  67.1648  | 0.4967 / 0.1231  |  0.2748 / 0.1681 |
> |  PRISM (vanilla)   |  5.5004  | 0.6985  / 0.8534  |  0.4864 / 0.6965 |
>
> |       CelebA    |     FID     |     P&R     |      D&C       |
> |:---:|:---:|:---:|:---:|
> |  PRISM       |  39.7997   | 0.6294 / 0.0713  |  0.4565 / 0.2967 |
> |  PRISM (vanilla)   |  19.1512  | 0.6621  / 0.3895  |  0.5348 / 0.5947 |
>
> **R1-2. Would the approach work for diffusion-based generative models? [R1-Q2]**
>
> Yes, adopting diffusion models (DDPM) [9] within PRISM is conceptually feasible. However, directly applying the MMD loss-based approach used in PRISM, which requires generating a set of samples for each update, poses practical challenges due to the well-known limitations of DDPM, including slow sampling and convergence rates. These limitations lead to significant computational costs, making the approach practically infeasible.
> Nevertheless, given the importance of diffusion models in generative modeling, we explored a modified version of our framework. Specifically, we retained the core concept of identifying SLT but replaced the MMD loss with diffusion loss to guide score updates. We conducted preliminary experiments on the MNIST dataset under a Non-IID, privacy-free scenario, and intermediate results are provided in Appendix H. While this approach is inherently sub-optimal—since it updates scores based on the loss of individual samples rather than the sample statistics—it still demonstrates the ability to generate recognizable shapes. With further tuning, we anticipate more promising results. However, as this lies beyond the scope of our current paper, it remains an interesting direction for future research. It is worth emphasizing that, to the best of our knowledge, this is the first federated generative model capable of generating data at this level of quality in a Non-IID, privacy-preserving setting for datasets more complex than MNIST or FMNIST. We hope the reviewers recognize the significance of this achievement.
>
> ...
>
> Please refer the remaining responses in the [Respones 2/2].

---

> ### Author Response · Authors · 2024-11-20
> **[Response 2/2] Thank you for constructive feedbacks!**
>
> **R1-3. Addressing the Concerns on Model and Dataset Complexity in Federated Learning
>  [R1-W1]**
>
> We would like to emphasize the significant challenges of training generative models under a federated learning (FL) setup. The field of federated generative models remains far behind the current trend of using more complex and powerful models, such as diffusion models, and large-scale datasets. This gap arises from the inherent difficulties in FL setups.
> Even for centralized differential privacy generative models [4-5], as also highlighted in the related work of our main script, training generative models is highly challenging and often unstable. Due to these obstacles, prior studies have primarily focused on simpler datasets, such as MNIST, and have been largely restricted to GANs. However, even under these conditions, many of these works failed to achieve desirable results. We summarize the FID scores and experimental setups reported in these works.
>
>
> |    FID, IID case  |    MD-GAN[6]   |    UA-GAN[7]   |   Multi-FLGAN[8]   |
> |:---:|:----------------------:|:--------------------:|:--------------------------:|
> |          MNIST      |         16.81       |         17.34       |         17.1       |
>
> While adapting our approach to models like diffusion models is beyond the scope of this paper (as noted in our response to R1-2, Appendix H), we recognize the importance of assessing performance on high-resolution and large-scale datasets.
> Considering computational resources and the rebuttal period, we conducted experiments on the CelebA 128x128 and CIFAR100 datasets under Non-IID conditions without considering privacy. The results are summarized in the table below, with qualitative results provided in Appendix.I. Please note that PRISM is the first to achieve this level of performance under the current setting, whereas existing baselines have struggled even on MNIST-level benchmarks.
>
> |    CelebA 128x128  |     FID      |     P&R      |      D&C       |
> |:---------------------------:|:--------------:|:---------------:|:--------------------:|
> |  PRISM       |  40.2927  |  0.7738 / 0.006  | 1.00207 / 0.348 |
>
> |       CIFAR100        |     FID     |     P&R     |      D&C       |
> |:---------------------------:|:--------------:|:---------------:|:---------------------:|
> |  PRISM       |  74.1609  | 0.6655 / 0.0719   |  0.602 / 0.3121 |
>
>
> [1] McMahan, Brendan, et al. "Communication-efficient learning of deep networks from decentralized data." Artificial intelligence and statistics. PMLR, 2017.
>
> [2] Zhao, Yue, et al. "Federated learning with non-iid data." arXiv preprint arXiv:1806.00582 (2018).
>
> [3] Hardy, Corentin, Erwan Le Merrer, and Bruno Sericola. "Md-gan: Multi-discriminator generative adversarial networks for distributed datasets." 2019 IEEE international parallel and distributed processing symposium (IPDPS). IEEE, 2019.
>
> [4] Dockhorn, Tim, et al. "Differentially private diffusion models." arXiv preprint arXiv:2210.09929 (2022).
>
> [5] Jiang, Zepeng, Weiwei Ni, and Yifan Zhang. "PATE-TripleGAN: Privacy-Preserving Image Synthesis with Gaussian Differential Privacy." arXiv preprint arXiv:2404.12730 (2024).
>
> [6] Hardy, Corentin, Erwan Le Merrer, and Bruno Sericola. "Md-gan: Multi-discriminator generative adversarial networks for distributed datasets." 2019 IEEE international parallel and distributed processing symposium (IPDPS). IEEE, 2019.
>
> [7] Zhang, Yikai, et al. "Training federated GANs with theoretical guarantees: A universal aggregation approach." arXiv preprint arXiv:2102.04655 (2021).
>
> [8] Amalan, Akash, et al. "Multi-flgans: multi-distributed adversarial networks for non-IID distribution." arXiv preprint arXiv:2206.12178 (2022).
>
> [9] Ho, Jonathan, Ajay Jain, and Pieter Abbeel. "Denoising diffusion probabilistic models." Advances in neural information processing systems 33 (2020): 6840-6851.

---

> > ### Author Response · Authors · 2024-11-24
> >
> > Dear Reviewer Aeqb,
> >
> > We sincerely thank you once again for the time and effort you have dedicated to reviewing this paper. Your invaluable feedback have significantly contributed to improving its quality.
> >
> > In the revised version, we have polished the manuscript, incorporated additional experimental results, and addressed your concerns with detailed clarifications. As the deadline for the Author-Reviewer discussion is approaching, we would like to ensure that our responses have sufficiently addressed your feedback. If there are any remaining concerns or additional clarifications or experiments that you would like us to provide, please do not hesitate to let us know.
> >
> > Thank you once again for your time and thoughtful input.
> >
> > Best regards,
> >
> > Paper 5404 Authors

---

> > > ### Author Response · Authors · 2024-12-01
> > > **Official Comment to Reviewer Aeqb. We are looking forward to your comment**
> > >
> > > Dear Reviewer Aeqb,
> > >
> > > With the author-reviewer discussion period nearing its conclusion, we kindly request your feedback on whether all your concerns have been fully addressed. Should you have any additional questions or require further clarification, please do not hesitate to let us know. We look forward to hearing from you.

---

### Meta-Review · Area_Chair_15hk · 2024-12-16

**Metareview:**

The paper proposes a federated masking method for generative models that reduces communication and memory costs as well as improves performance in non-iid settings. The authors have addressed the comments and questions from the reviewers. None of the reviewers against the acceptance of the paper. Please revise and incorporate all the discussions into the final version.

**Additional Comments On Reviewer Discussion:**

NA

---

### Decision · Program_Chairs · 2025-01-22

Accept (Poster)